# Temporal transcriptional response of *Candida glabrata* during macrophage infection reveals a multifaceted transcriptional regulator CgXbp1 important for macrophage response and fluconazole resistance

**Maruti Nandan Rai**[1†]**, Qing Lan**[1†]**, Chirag Parsania**[1]**, Rikky Rai**[1]**, Niranjan Shirgaonkar**[1]**, Ruiwen Chen**[1]**, Li Shen**[1,2]**, Kaeling Tan**[1,2]**, Koon Ho Wong**[1,3,4]*****

[1]Faculty of Health Sciences, University of Macau, Taipa, China; [2]Gene Expression, Genomics and Bioinformatics Core, Faculty of Health Sciences, University of Macau, Taipa, China; [3]Institute of Translational Medicine, Faculty of Health Sciences, University of Macau,Avenida da Universidade, Taipa, China; [4]MoE Frontiers Science Center for Precision Oncology, University of Macau, Taipa, China

**\*For correspondence:**
koonhowong@um.edu.mo

[†]These authors contributed equally to this work

**Competing interest:** The authors declare that no competing interests exist.

**Abstract** *Candida glabrata* can thrive inside macrophages and tolerate high levels of azole anti-fungals. These innate abilities render infections by this human pathogen a clinical challenge. How *C. glabrata* reacts inside macrophages and what is the molecular basis of its drug tolerance are not well understood. Here, we mapped genome-wide RNA polymerase II (RNAPII) occupancy in *C. glabrata* to delineate its transcriptional responses during macrophage infection in high temporal resolution. RNAPII profiles revealed dynamic *C. glabrata* responses to macrophages with genes of specialized pathways activated chronologically at different times of infection. We identified an uncharacterized transcription factor (CgXbp1) important for the chronological macrophage response, survival in macrophages, and virulence. Genome-wide mapping of CgXbp1 direct targets further revealed its multi-faceted functions, regulating not only virulence-related genes but also genes associated with drug resistance. Finally, we showed that CgXbp1 indeed also affects fluconazole resistance. Overall, this work presents a powerful approach for examining host-pathogen interaction and uncovers a novel transcription factor important for *C. glabrata*'s survival in macrophages and drug tolerance.

## Editor's evaluation

This important study reveals, with exquisite temporal resolutions, critical transcriptional events that take place as Candida glabrata infects macrophages, providing convincing analyses that enhance our current understanding of the underlying sequential transcriptional changes, including a previously uncharacterized transcription factor (CgXbp1), which plays an important role in modulating the temporal responses in macrophages, impacting C. glabrata survival and virulence and, notably, also fluconazole resistance. The work would benefit from additional experiments that could provide a more mechanistic understanding of the key events leading to successful infection yet, in its current form it should be of interest to a broad audience interested in host-pathogen interactions, fungal biology, and transcriptional mechanisms at large.

## Introduction

Phagocytes such as macrophages constitute the first line of host immune defence against invading pathogens (*Brown, 2011*; *Erwig and Gow, 2016*). The ability to escape or survive phagocytic attacks is fundamental to the virulence of pathogens (*Erwig and Gow, 2016*; *Seider et al., 2010*). *Candida* species are prominent opportunistic fungal pathogens with an associated mortality rate of ~30–60% among immunocompromised populations (*Bongomin et al., 2017*; *Lamoth et al., 2018*). *Candida albicans* is responsible for most Candidiasis, although recent studies indicate an epidemiological shift in Candidiasis with an upsurge in infections caused by *Candida glabrata* (*Benedict et al., 2017*; *Katsipoulaki et al., 2024*; *Lamoth et al., 2018*), which has recently been renamed as *Nakaseomyces glabratus* (*Takashima and Sugita, 2022*). Relative to other fungal species including *C. albicans*, *C. glabrata* is more resistant to antifungal drugs like fluconazole and can survive and proliferate inside immune cells (*Katsipoulaki et al., 2024*; *Rai et al., 2012*; *Seider et al., 2011*). Thus far, details about how *C. glabrata* survives, adapts, and proliferates in phagocytes and the basis for its intrinsically high azole resistance are still not clearly understood.

Genome-wide transcriptomic studies have been performed to map the response of *Candida species* during macrophage infection (*Kaur et al., 2007*; *Lorenz et al., 2004*; *Lorenz and Fink, 2001*; *Rai et al., 2012*; *Rubin-Bejerano et al., 2003*), but the insights gained into the infection process so far lack temporal resolution, centering mostly on the late stages of the pathogen-host interactions. We reason that the immediate and early pathogen response is pivotal for survival and adaptation in the host, while responses during later stages reflect strategies for growth and proliferation. Therefore, delineating the whole episode of pathogen response, instead of just a snapshot, during infection is fundamental to understanding pathogenesis. However, conventional transcriptomic analysis involving mRNAs are less suitable for dissecting dynamic temporal transcriptional changes, as measurements of mRNA levels are convoluted by transcript stabilities (*Tan and Wong, 2019*).

Here, we applied the powerful Chromatin Immuno-precipitation followed by the Next Generation Sequencing (ChIP-seq) method against elongating RNA Polymerase II (RNAPII) to map *C. glabrata* transcriptional responses during macrophage infection. We show that *C. glabrata* responds to macrophage infection by mounting chronological transcriptional responses. Based on the transcription pattern, we identified many candidate transcriptional regulators including a novel transcription factor, CgXbp1, for the macrophage response. Deletion of *CgXBP1* led to accelerated transcriptional activation of genes associated with multiple biological processes during interaction with macrophages. We further demonstrate that CgXbp1 is a multifaceted transcription factor directly binding to many *C. glabrata* genes with functions associated with the pathogenesis and drug resistance processes. *CgXBP1* deletion resulted in attenuated survival in host macrophages, diminished virulence in the *Galleria mellonella* model of Candidiasis, and elevated resistance to the antifungal drug fluconazole. Overall, our work uncovers an important novel transcription factor for *C. glabrata*'s survival in macrophages and antifungal drug resistance.

## Results

### Mapping high temporal resolution transcriptional responses of *C. glabrata* during macrophage infection

To understand how *C. glabrata* survives macrophage phagocytosis, we applied ChIP-seq against elongating RNAPII in a time-course experiment after 0.5, 2, 4, 6, and 8 hr of THP-1 macrophage infection to map genome-wide transcription responses of *C. glabrata* during different stages of THP-1 macrophage infection (*Figure 1A*). As expected, genes known to be induced by macrophage phagocytosis e.g., tricarboxylic acid [TCA] cycle, glyoxylate bypass, and iron homeostasis genes (*Kaur et al., 2007*; *Rai et al., 2012*) had significant RNAPII occupancies at their gene bodies specifically but not at inter-genic regions (*Figure 1B*, *Figure 1—figure supplement 1A*). In addition, the ChIP-seq data also revealed temporal gene expression information. For example, the ATP synthesis gene *CgCYC1* was dramatically up-regulated immediately (0.5 hr) upon macrophage internalisation, while the TCA cycle gene *CgCIT2* and glyoxylate bypass gene *CgICL1* were induced slightly later at 2 hr and their transcription levels decreased subsequently (4–6 hr *Figure 1B*). In contrast, an opposite transcription pattern (e.g. gradual increasing and peaking at later stages) was observed for *CgFTR1*, *CgTRR1*, and *CgMT-I*, which are involved in iron uptake, oxidative stress response, and sequestration of metal

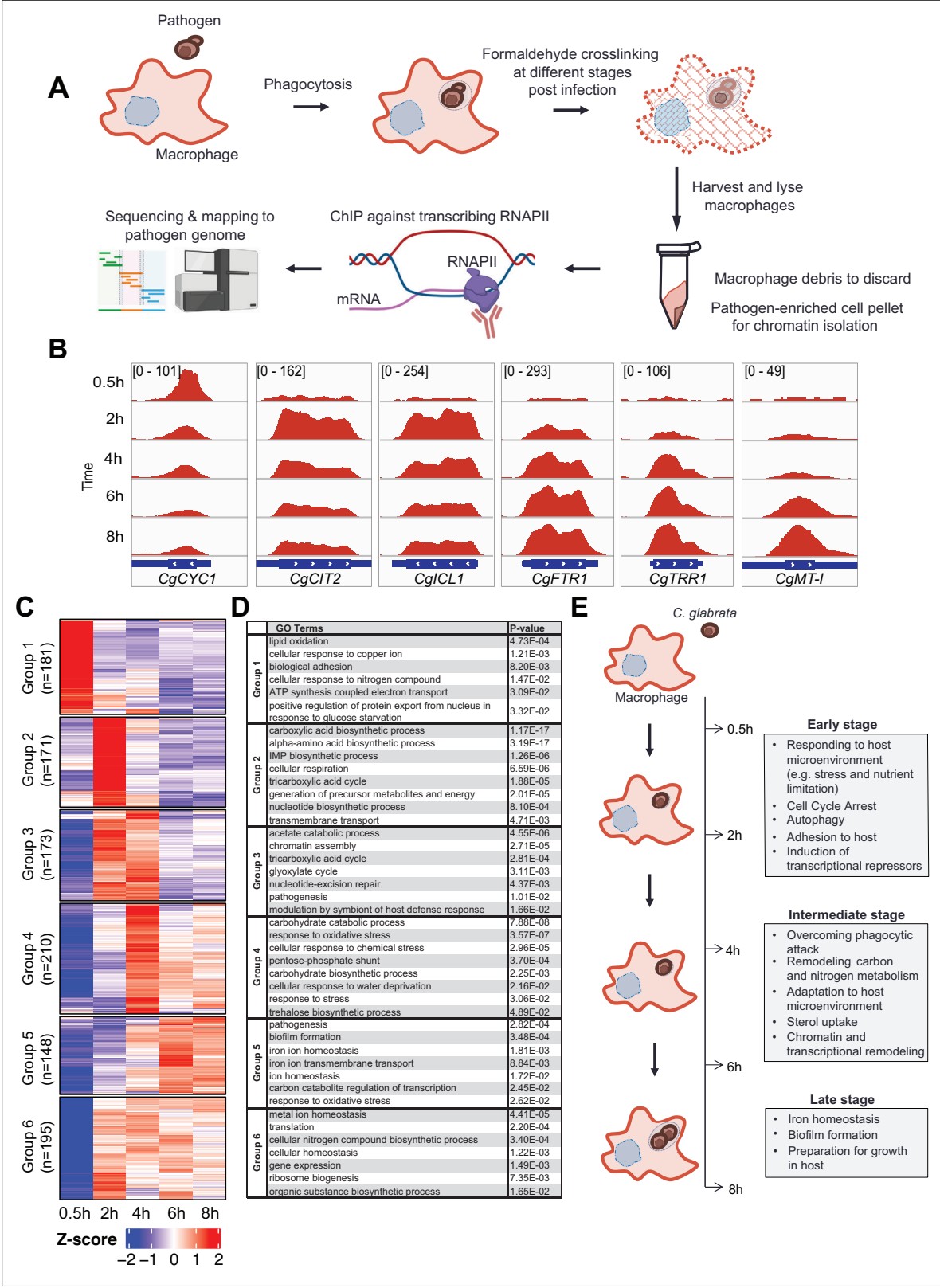

**Figure 1.** *C. glabrata* mounts a dynamic chronological transcriptional response upon macrophage infection. (**A**) A schematic diagram showing the overall methodology used in this study. (**B**) Genome browser views of RNA polymerase II (RNAPII) Chromatin Immuno-precipitation followed by the Next Generation Sequencing (ChIP-seq) signals on *CgCYC1*, *CgCIT2*, *CgICL1*, *CgFTR1*, *CgTRR1*, and *CgMT-I* genes at indicated time points. Numbers in the square brackets indicate the y-axis scale range of normalized RNAPII ChIP-seq signal used for the indicated genes across different datasets.

*Figure 1 continued on next page*

*Figure 1 continued*

(**C**) A heatmap showing temporal expression patterns of transcribed genes in *C. glabrata* during 0.5–8 hr macrophage infection in a time-course experiment. The colour scale represents the Z-score of the normalized RNAPII ChIP-seq signal. The groups were determined by *k*-means clustering. (**D**) A table showing significantly enriched gene ntology o(GO) biological processes (p-value ≤0.05) for the six groups of temporally transcribed genes. (**E**) A schematic diagram showing *C. glabrata* transcriptional responses (broadly classified into early, intermediate, and late stages) during macrophage infection.

The online version of this article includes the following figure supplement(s) for figure 1:

**Figure supplement 1.** High-resolution RNA polymerase II (RNAPII) Chromatin Immuno-precipitation followed by the Next Generation Sequencing (ChIP-seq) can capture genome-wide active and temporally induced transcription activities in *C. glabrata* during macrophage infection.

**Figure supplement 2.** *C. glabrata* undergoes cell cycle arrest upon macrophage phagocytosis.

**Figure supplement 3.** Virulence-centric biological processes are temporally activated in *C.glabrata* at different stages of macrophage infection.

**Figure supplement 4.** Genes encoding histones or proteins involved in glycolysis, gluconeogenesis pathways, and chromatin modification and remodelling are transcriptionally induced in *C. glabrata* during the macrophage infection.

**Figure supplement 5.** Tip1-related (TIR) family genes for sterol uptake displaying co-expression during macrophage infection.

**Figure supplement 6.** Correlation between independent biological repeats of the RNA polymerase II (RNAPII) Chromatin Immuno-precipitation followed by the Next Generation Sequencing (ChIP-seq) experiment for wild-type and the *Cgxbp1Δ* mutant.

ions, respectively (*Figure 1B*). Therefore, the RNAPII ChIP-seq approach can capture temporal gene expression changes in *C. glabrata* during macrophage infection.

## *C. glabrata* mounts dynamic, temporal, and chronological transcription responses during macrophage infection

Systematic analysis of actively transcribed genes revealed that approximately 30% of *C. glabrata* genes (n=1589, *Supplementary file 1*) were constitutively transcribed (n=511, *Figure 1—figure supplement 1B*) or temporally induced (n=1078, *Figure 1—figure supplement 1C*) during macrophage infection. Temporally induced genes were further classified into six groups according to their transcription pattern using *k*-means clustering. The overall transcriptional response was highly diverse with each group exhibiting a unique temporal transcriptional pattern (*Figure 1C*). Interestingly, while some genes were induced immediately (0.5 hr, Group 1, n=181) upon internalization by macrophages, transcriptional induction of over 80% of genes (Group 2–6, n=897) did not happen until later (2–8 hr). Besides, their expression patterns were highly variable, illustrating the complex and dynamic nature of *C. glabrata* transcriptional response during macrophage infection.

Gene Ontology (GO) analysis revealed the chronological activation of different biological processes during the infection process (*Figure 1D*, *Supplementary file 2*). In the immediate response (0.5 hr), genes (Group 1, n=181) were significantly enriched in processes such as adhesion, responses to copper ion and nitrogen compound, positive regulation of nuclear export in response to glucose starvation, lipid oxidation, and ATP synthesis (*Figure 1D*). This indicated that *C. glabrata* experiences nutrient and energy deprivation immediately upon entry to macrophages. Alternatively, the induction of ATP biosynthesis genes may reflect a strong demand for energy by *C. glabrata* to deal with the host's attacks and/or to adapt to the host microenvironment. Subsequently (2 hr post phagocytosis), *C. glabrata* underwent a major metabolic remodelling presumably to prepare for growth and generate energy, as reflected by the next wave of transcriptional induction for genes (Group 2, n=171) involved in the TCA cycle, biosynthesis of inosine 5' monophosphate (IMP), carboxylic acid, amino acid, nucleotide, and precursor for metabolite and energy (*Figure 1C and D*). In addition, cell cycle arrest and DNA damage checkpoint genes (*CgMEC3, CgGLC7, CAGL0G07271g, and CAGL0A04213g*) were also strongly induced at this early stage (*Figure 1—figure supplement 2A*), and *C. glabrata* cells were indeed arrested at the G1-S phase cell cycle after macrophage engulfment (*Figure 1—figure supplement 2B*). It is noteworthy that many genes and pathways previously shown to be critical for *C. glabrata* virulence (*Kasper et al., 2015*; *Kaur et al., 2005*; *Rai et al., 2012*) such as adherence, response to DNA damage, oxidative stress, autophagy, TCA cycle, amino acid biosynthesis, and iron homeostasis were markedly induced at the early stages (0.5 and 2 hr *Figure 1—figure supplement 3A–G*). Therefore, virulence-centric biological processes were among the most immediate *C. glabrata* responses upon macrophage phagocytosis, implying the importance of the early transcriptional response towards its adaptation and survival in macrophages.

During the next stage of infection (2–4 hr), *C. glabrata* continued to actively transcribe genes associated with carbon metabolism, DNA repair, and pathogenesis (Group 3, *Figure 1C and D*). It is interesting to note that gluconeogenesis and glycolysis genes (as determined based on the genes of the unidirectional rate-limiting steps) were both induced during macrophage infection but at different times – first with gluconeogenesis genes followed by glycolysis genes (*Figure 1—figure supplement 4A*), suggesting that phagocytosed *C. glabrata* cells were still trying to achieve metabolic homeostasis and to counter macrophage internal milieu at this stage. In addition, genes required for chromatin assembly and modification were also significantly induced at this stage (*Figure 1—figure supplement 4B and C*), supporting an earlier report about the involvement of chromatin remodelling during the infection process (*Rai et al., 2012*). Towards the later phase of this stage (4 hr), genes for responses to different stresses (e.g. oxidative, chemical stress, and osmolarity) and resistance thermo-tolerance and oxidative stress (e.g. trehalose biosynthesis and pentose phosphate pathway, respectively) become maximally induced (Group 4, *Figure 1C and D*). The induction of these stress response pathway genes towards the end of metabolic remodelling is somewhat unexpected, as it suggests that phagocytic attacks (e.g. ROS) against *C. glabrata* might not have occurred until the later phase. However, as shown above, the observation that DNA repair and damage response genes were already upregulated at 2 hr indicates that cells had already experienced the attacks. These findings collectively suggest that *C. glabrata* elicits a coordinated stage-wise response during infection; first adapting to macrophage nutrient microenvironment before overcoming phagocytic attacks. Interestingly, a family of sterol uptake genes (known as *TIR* [Tip1-related]) displayed concerted transcription activation at the end of this stage (4 hr *Figure 1—figure supplement 5*). In *Saccharomyces cerevisiae*, *TIR* genes are activated and required for growth under anaerobic conditions (*Abramova et al., 2001*). Given that sterols are an essential component of the cell membrane and that ergosterol biosynthesis is an oxygen-dependent process (*Joffrion and Cushion, 2010*), the up-regulation of the *TIR* genes indicates an experience of oxygen deprivation and a need for sterols (presumably for proliferation) by *C. glabrata*.

Towards the late stage (6–8 hr), genes required for biofilm formation, and iron homeostasis, both of which play critical roles in the pathogenesis process (*Rodrigues et al., 2017*; *Seider et al., 2014*), became maximal induced (Group 5, *Figure 1C and D*). In fact, over 70% of the previously identified iron-responsive genes (154 out of 214, *Supplementary file 3*; *Denecker et al., 2020*) were induced during macrophage infection. As iron homeostasis is necessary for cell growth and proliferation, this observation potentially suggests that the cells are preparing for growth, and this is consistent with the concomitant induction of the biofilm formation genes that are also necessary for proliferation. Altogether, the overall results revealed details of the dynamic stage-wise responses of *C. glabrata* during macrophage infection (*Figure 1E*).

## Identification of potential transcriptional regulators of early temporal response

We next attempted to identify the potential transcriptional regulators for the chronological transcriptional response. Remarkably, more than 25% of *C. glabrata* transcription factor (TF) genes (n=53) were expressed during macrophage infection (*Table 1*), with 39 TF genes showing a temporal induction pattern (*Figure 2A*). Of note, eleven TFs (Aft1, Ap5, Haa1, Hap4, Hap5, Msn4, Upc2, Yap1, Yap3, Yap6, and Yap7) are known to either bind or control some of the macrophage infection-induced genes (*Supplementary file 4*) as reported by PathoYeastract (*Monteiro et al., 2020*). More importantly, these TFs are known to control genes for response to and survival inside macrophages; e.g., Aft1, Hap4, Yap1, and Yap7 have been shown to regulate iron homeostasis genes (*Denecker et al., 2020*; *Denecker et al., 2020*; *Merhej et al., 2016*), Msn4 and Yap1 regulate oxidative stress response (*Cuéllar-Cruz et al., 2008*; *Roetzer et al., 2010*) and Yap6 is important for pH stresses (*Zhou et al., 2020*). These provide strong support to the identified TFs being responsible for the observed temporal transcriptional response.

As the early response is likely to have influential effects on infection outcome, we focused on the four candidate TFs in Group 1 (induction at 0.5 hr); the genes *CAGL0F00561g*, *CAGL0G02739g*, *CAGL0L03157g*, and *CAGL0J04400g* are uncharacterized and annotated as the *Saccharomyces cerevisiae* orthologue of *RPA12*, *XBP1*, *DAL80*, and *HAP3*, respectively. Interestingly, three of the four yeast orthologues (*RPA12*, *XBP1*, *DAL80*) have been described as repressors (*Mai and Breeden*,

**Table 1.** Constitutively transcribed or temporally induced *C. glabrata* transcription factor genes during macrophage infection.

**Temporally induced**

| Group number | Cg common name | Cg ORF name | Sc common name | Sc gene desc |
|---|---|---|---|---|
| Group:1:(n=4) | CAGL0G02739g | CAGL0G02739g | XBP1 | XhoI site-Binding Protein |
| Group:1:(n=4) | CAGL0L03157g | CAGL0L03157g | DAL80 | Degradation of Allantoin |
| Group:1:(n=4) | CAGL0J04400g | CAGL0J04400g | HAP3 | Heme Activator Protein |
| Group:1:(n=4) | CAGL0F00561g | CAGL0F00561g | RPA12 | RNA Polymerase A |
| Group:2:(n=4) | CAGL0K06413g | CAGL0K06413g | STP1 | Species-specific tRNA Processing |
| Group:2:(n=4) | CAGL0E00737g | CAGL0E00737g | HMO1 | High MObility group (HMG) family |
| Group:2:(n=4) | MET28 | CAGL0K08668g | MET28 | METhionine |
| Group:2:(n=4) | CAGL0J03608g | CAGL0J03608g | HCM1 | High-Copy suppressor of Calmodulin |
| Group:3:(n=5) | RTG1 | CAGL0C05335g | RTG1 | ReTroGrade regulation |
| Group:3:(n=5) | CAGL0J01177g | CAGL0J01177g | ABF1 | ARS-Binding Factor 1 |
| Group:3:(n=5) | CAGL0K04543g | CAGL0K04543g | SPT4 | SuPpressor of Ty's |
| Group:3:(n=5) | HAP4 | CAGL0K08624g | HAP4 | Heme Activator Protein |
| Group:3:(n=5) | CAGL0G07249g | CAGL0G07249g | YHP1 | Yeast Homeo-Protein |
| Group:4:(n=4) | CAGL0L07480g | CAGL0L07480g | NRG2 | Negative Regulator of Glucose-controlled genes |
| Group:4:(n=4) | MIG1 | CAGL0A01628g | MIG1 | Multicopy Inhibitor of GAL gene expression |
| Group:4:(n=4) | CAGL0G08646g | CAGL0G08646g | POG1 | Promoter Of Growth |
| Group:4:(n=4) | CAGL0K02145g | CAGL0K02145g | COM2 | Cousin of Msn2 |
| Group:5:(n=17) | RME1 | CAGL0K04257g | RME1 | Regulator of MEiosis |
| Group:5:(n=17) | CAGL0M07634g | CAGL0M07634g | SOK2 | Suppressor Of Kinase |
| Group:5:(n=17) | CAGL0M01716g | CAGL0M01716g | TEC1 | Transposon Enhancement Control |
| Group:5:(n=17) | CAGL0F07909g | CAGL0F07909g | TBS1 | ThiaBendazole Sensitive |
| Group:5:(n=17) | UPC2B | CAGL0F07865g | UPC2 | UPtake Control |
| Group:5:(n=17) | ZAP1 | CAGL0J05060g | ZAP1 | Zinc-responsive Activator Protein |
| Group:5:(n=17) | CAGL0C02519g | CAGL0C02519g | MIG3 | Multicopy Inhibitor of Growth |
| Group:5:(n=17) | HAP5 | CAGL0K09900g | HAP5 | Heme Activator Protein |
| Group:5:(n=17) | CAGL0E04312g | CAGL0E04312g | STP2 | protein with similarity to Stp1p |
| Group:5:(n=17) | CAGL0B03421g | CAGL0B03421g | HAP1 | Heme Activator Protein |
| Group:5:(n=17) | HAA1 | CAGL0L09339g | HAA1 | Homolog of Ace1 Activator |
| Group:5:(n=17) | GAT1 | CAGL0K07634g | GAT1 | Transcriptional activator of nitrogen catabolite repression genes |
| Group:5:(n=17) | YAP6 | CAGL0M08800g | YAP6 | Yeast homolog of AP-1 |
| Group:5:(n=17) | GLM6 | CAGL0J01595g | #N/A | #N/A |
| Group:5:(n=17) | AFT1 | CAGL0H03487g | AFT1 | Activator of Ferrous Transport |
| Group:5:(n=17) | YAP3b | CAGL0M10087g | #N/A | #N/A |
| Group:5:(n=17) | CAGL0E03762g | CAGL0E03762g | RIM101 | Regulator of IME2 |
| Group:6:(n=5) | AP5 | CAGL0K08756g | YAP5 | Yeast AP-1 |
| Group:6:(n=5) | GCN4 | CAGL0L02475g | GCN4 | General Control Nonderepressible |
| Group:6:(n=5) | CAGL0E05566g | CAGL0E05566g | TYE7 | Ty1-mediated Expression |

*Table 1 continued on next page*

*Table 1 continued*

**Temporally induced**

| Group | Cg common name | Cg ORF name | Sc common name | Sc gene desc |
|---|---|---|---|---|
| Group:6:(n=5) | *RPN4* | *CAGL0K01727g* | *RPN4* | Regulatory Particle Non-ATPase |
| Group:6:(n=5) | *CAGL0C01551g* | *CAGL0C01551g* | *TOS8* | Target Of Sbf |

**Constitutively transcribed**

| Group | Cg common name | Cg ORF name | Sc common name | Sc gene desc |
|---|---|---|---|---|
| Constitutively transcribed | *PHO2* | *CAGL0L07436g* | *PHO2* | PHOsphate metabolism |
| Constitutively transcribed | *AP1* | *CAGL0H04631g* | *YAP1* | Yeast AP-1 |
| Constitutively transcribed | *CAGL0M04983g* | *CAGL0M04983g* | *MBF1* | Multiprotein Bridging Factor |
| Constitutively transcribed | *MSN4* | *CAGL0M13189g* | *MSN4* | Multicopy suppressor of SNF1 mutation |
| Constitutively transcribed | *CAGL0E00891g* | *CAGL0E00891g* | *STB3* | Sin Three Binding protein |
| Constitutively transcribed | *CAD1* | *CAGL0F03069g* | *CAD1* | CADmium resistance |
| Constitutively transcribed | *CAGL0A04257g* | *CAGL0A04257g* | *TOD6* | Twin Of Dot6p |
| Constitutively transcribed | *CAGL0I08635g* | *CAGL0I08635g* | *BUR6* | Bypass UAS Requirement |
| Constitutively transcribed | *YAP7* | *CAGL0F01265g* | *YAP7* | Yeast AP-1 |
| Constitutively transcribed | *CAGL0L02013g* | *CAGL0L02013g* | *IXR1* | Intrastrand cross (X)-link Recognition |
| Constitutively transcribed | *CAGL0M01474g* | *CAGL0M01474g* | *NCB2* | Negative Cofactor B |
| Constitutively transcribed | *CAGL0F06259g* | *CAGL0F06259g* | *ARG80* | ARGinine requiring |
| Constitutively transcribed | *SWI5* | *CAGL0E01331g* | *SWI5* | SWItching deficient |
| Constitutively transcribed | *CAGL0M09955g* | *CAGL0M09955g* | *SFP1* | Split Finger Protein |

*1997*; *Marzluf, 1997*; *Yadav et al., 2016*). Transcription regulatory network analysis by PathoYeastract (*Monteiro et al., 2020*) further showed that the orthologues of ~35% macrophage infection-induced genes are targets of Xbp1 in *S. cerevisiae* (n=375 out of 1078, respectively; *Figure 2—figure supplement 1A*, *Supplementary file 5*), including a significant number of TF genes (n=14; *Figure 2—figure supplement 1B*). In contrast, a much smaller set of orthologous genes (~7%, n=72, *Figure 2—figure supplement 1A*) is annotated as being *S. cerevisiae* Hap3 targets, while no information was available on the PathoYeastract database (*Monteiro et al., 2020*) for the other two repressors. These results suggest that the chronological transcriptional response upon macrophage phagocytosis involves the interplays between transcriptional repressors and activators and that the protein encoded by *CAGL0G02739g* (hereafter referred to as *CgXBP1*) likely plays a central role in orchestrating the overall response.

## The transcription factor CgXbp1 binds to the promoter of many early temporal response genes

To further characterize the role of CgXbp1 during macrophage infection, we tagged CgXbp1 with the MYC epitope, and examined its levels before and after macrophage infection by Western blot analysis. Consistent with the RNAPII result, CgXbp1[MYC] protein was expressed at a low level before macrophage infection and was significantly induced upon macrophage internalization (*Figure 2B*).

ChIP-seq was performed to identify CgXbp1[MYC] genome-wide targets before and after macrophage phagocytosis. ChIP against CgXbp1[MYC] before macrophage infection failed to pull down sufficient DNA material for sequencing library preparation, presumably due to the low CgXbp1[MYC] expression (*Figure 2B*). In contrast, many distinct CgXbp1[MYC] ChIP-seq peaks were detected throughout the genome in macrophage-phagocytosed cells (*Figure 2C*). A total of 251 CgXbp1[MYC] binding sites were commonly identified in biological replicates by MACS2 (Model-based Analyses for ChIP-seq) peak-calling analysis (*Zhang et al., 2008*; *Figure 2—figure supplement 2A*, *Supplementary file 6*). The

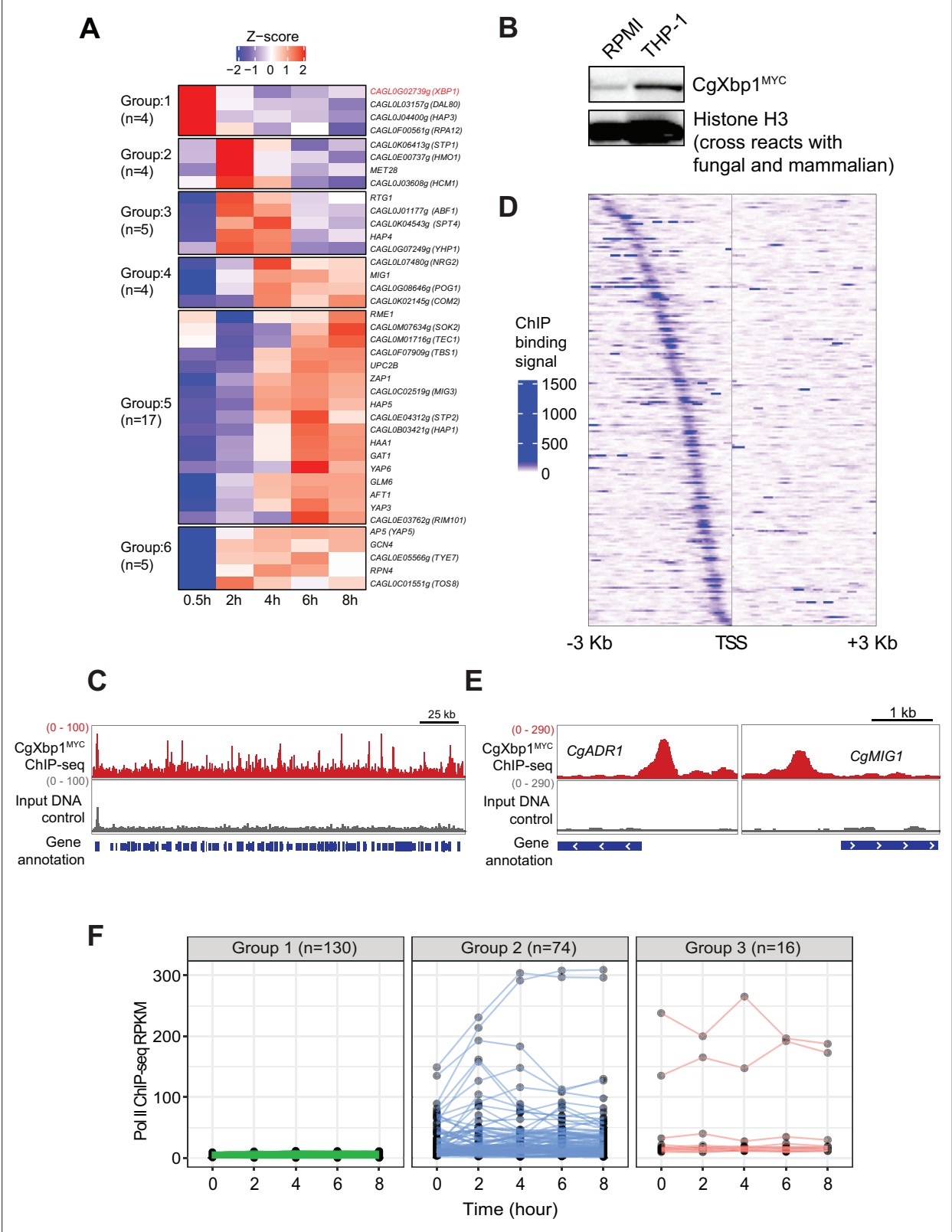

**Figure 2.** CgXbp1 is central in orchestrating the dynamic transcriptional response of *C.glabrata* during macrophage infection. (**A**) A heatmap showing temporal expression patterns of *C. glabrata* transcription factor genes transcribed during THP-1 macrophage infection. Color scale represents the Z-score of the normalized RNAPII ChIP-seq signal. The groups of temporally induced genes were determined by *k*-means clustering. (**B**) Western blot analysis of CgXbp1 expression during THP1 macrophage infection. (**C**) Representative genome-browser screenshots showing CgXbp1MYC ChIP-seq

*Figure 2 continued on next page*

*Figure 2 continued*

signal on a chromosomal region. (**D**) A Heat map of ChIP-seq signals on promoters of CgXbp1 target genes. The colour scale indicates normalized ChIP-seq signal on 3 kb upstream and downstream flanking regions from the transcription start site (TSS) of the target genes. (**E**) Representative genome-browser screenshots showing CgXbp1^MYC ChIP-seq signal on the promoters of *CgMIG1* and *CgADR1*. (**F**) CgXbp1 target genes displaying RNAP II binding signal at indicated time points during macrophage infection. The groups were classified based on gene expression patterns. Group 1 includes minimally transcribed genes with FPKM values less than 12. Group 2 contains the genes with FPKM values greater than 12 and a highly variable expression pattern (fold change between maximum and minimum is greater than 1.5). Group 3 involves the genes with FPKM greater than 12 but less variable expression levels (fold change between maximum and minimum is less than 1.5).

The online version of this article includes the following source data and figure supplement(s) for figure 2:

**Source data 1.** Original files for the western blots shown in *Figure 2B*.

**Source data 2.** A Microsoft Word file containing original western blots for *Figure 2B*, indicating the relevant bands and treatments.

**Figure supplement 1.** CgXbp1 is a key transcription regulator of the temporal transcriptional response of *C.glabrata* during macrophage infection.

**Figure supplement 2.** CgXbp1^MYC ChIP-seq binding signals during macrophage infection.

peaks were located at the promoter of 220 genes (*Figure 2D*, *Supplementary file 7*), of which 48% of them were up-regulated during macrophage infection. In *S. cerevisiae*, Xbp1 directly represses the expression of the *CLN3* gene, which encodes the G1 cyclin, to arrest cells at the G1 phase of the cell cycle (*Miles et al., 2013*). The *CLN3* gene in *C. glabrata* is not listed in the 220 CgXbp1 ChIP-seq target genes, but an Xbp1 binding site was present at approximately 1.2 kb upstream of *CLN3* promoter separated by a dubious annotated gene (*CAGL0M12001g* *Figure 2—figure supplement 2B*), which is predicted to encode a small protein of 86 amino acids with no orthologous sequence in any fungal species based on Blast analysis. It is possible that the gene is a wrong annotation and that the upstream Xbp1 binding site may be controlling *CLN3* expression, like in *S. cerevisiae*, although there was no difference in the transcription levels (RNAPII occupancy) of *CLN3* between the *Cgxbp1Δ* mutant and wild-type during the infection time course (*Figure 2—figure supplement 2C*).

GO analysis showed that these CgXbp1 target genes were significantly associated with major biological processes such as 'regulation of transcription,' 'transmembrane transport,' 'response to copper ion,' 'development of symbiont,' 'carbohydrate metabolic process,' 'pseudohyphal growth,' and 'biofilm formation' (*Table 2*, *Supplementary file 8*). Some of these processes are important for host infection. These functions are also consistent with the above findings that CgXbp1 is important for *C. glabrata* response and survival in macrophages. More importantly, CgXbp1^MYC target genes include 27 transcription regulators (*Figure 2E*, *Supplementary file 9*), implying that CgXbp1 also indirectly controls many other pathways by regulating the hierarchy of different gene regulatory networks. Of note, most of the regulator genes have not been characterized in *C. glabrata*, while their *S. cerevisiae* orthologues regulate genes of diverse physiological pathways that are important for virulence, such as carbon metabolism (Mig1, Adr1, Rgm1, and Tye7), nitrogen metabolism (Gat2), amino acid biosynthesis (Leu3), DNA damage (Rfx1 and Imp21), pH response (Rim101) and pseudohyphal formation (Phd1 and Ste12 *Supplementary file 9*).

Notably, more than half of the CgXbp1-bound genes (130 out of 220) were minimally transcribed (i.e. they have background levels of RNAPII ChIP-seq signal), if any, during macrophage infection (*Figure 2F*), presumably their transcription activators are not expressed or functional under the condition. Most of the remaining genes (74 out of 90 genes) had low expression in wild-type *C. glabrata* during the early stage of macrophage infection when CgXbp1 expression is at the highest level, while their expression was temporally induced subsequently (Group 2 in *Figure 2F*), suggesting that CgXbp1 represses their expression during the early infection stage.

## CgXbp1 is crucial for the chronological transcriptional response during macrophage infection

We next deleted the *CgXBP1* gene and analyzed the transcriptional response of the *Cgxbp1Δ* mutant to macrophages. RNAPII ChIP-seq time course analysis showed that a similar number of genes were transcribed in the mutant during macrophage infection (1471 vs 1589 genes in *Cgxbp1Δ* and wild-type, respectively) (*Supplementary file 10*) and ~90% of the transcribed genes are common between wild-type and the mutant (*Figure 3—figure supplement 1A*), suggesting that CgXbp1 has little effect on the overall set of genes transcribed during macrophage infection. Nevertheless, there are 295

**Table 2.** Table of significantly enriched and non-redundant gene ontology (GO)-terms for biological processes among CgXbp1 target genes during macrophage infection.

| GO-term for biological processes | p-value | Genes in the background | CgXbp1 bound genes |
|---|---|---|---|
| regulation of transcription, DNA-templated | 0.0012 | 468 | 33 |
| transmembrane transport | 0.0001 | 302 | 27 |
| positive regulation of transcription, DNA-templated | 0.0007 | 257 | 22 |
| carbohydrate metabolic process | 0.0029 | 215 | 18 |
| cellular carbohydrate metabolic process | 0.013 | 125 | 11 |
| negative regulation of transcription, DNA-templated | 0.0448 | 151 | 11 |
| regulation of filamentous growth | 0.0455 | 133 | 10 |
| cCarbohydrate catabolic process | 0.0074 | 56 | 7 |
| iInterspecies interaction between organisms | 0.019 | 67 | 7 |
| polysaccharide biosynthetic process | 0.0156 | 50 | 6 |
| regulation of cell growth | 0.0156 | 50 | 6 |
| positive regulation of pseudohyphal growth | 0.0013 | 21 | 5 |
| pyruvate metabolic process | 0.0068 | 30 | 5 |
| regulation of pseudohyphal growth | 0.0148 | 36 | 5 |
| regulation of carbohydrate metabolic process | 0.0356 | 45 | 5 |
| sphingolipid metabolic process | 0.0419 | 47 | 5 |
| development of symbiont in host | 0.002 | 14 | 4 |
| response to copper ion | 0.0043 | 17 | 4 |
| cellular glucose homeostasis | 0.0095 | 21 | 4 |
| glycolytic process | 0.0112 | 22 | 4 |
| nucleoside diphosphate phosphorylation | 0.0153 | 24 | 4 |
| nucleotide phosphorylation | 0.023 | 27 | 4 |
| sphingolipid biosynthetic process | 0.0443 | 33 | 4 |
| (1->3)-beta-D-glucan biosynthetic process | 0.0066 | 10 | 3 |
| glutamate metabolic process | 0.0114 | 12 | 3 |
| regulation of Rho protein signal transduction | 0.0143 | 13 | 3 |
| transfer RNA gene-mediated silencing | 0.0143 | 13 | 3 |
| glucose-mediated signaling pathway | 0.0177 | 14 | 3 |
| chromatin silencing by small RNA | 0.0256 | 16 | 3 |
| Rho protein signal transduction | 0.0302 | 17 | 3 |
| response to glucose | 0.0352 | 18 | 3 |
| cellular response to carbohydrate stimulus | 0.0352 | 18 | 3 |

and 177 genes with detectable transcription only in wild-type or the *Cgxbp1Δ* mutant, respectively (*Figure 3—figure supplement 1A*; *Supplementary file 10*). Notably, the *Cgxbp1Δ* mutant had a significantly higher number of genes activated at the earliest infection time point (0.5 hr, *Figure 3A*, *Supplementary file 10*) as compared to wild-type (*Figure 1C*); e.g., 369 genes showed accelerated expression in the *Cgxbp1Δ* mutant, while 162 and 109 genes had an unchanged or delayed gene expression profile (*Figure 3B*).

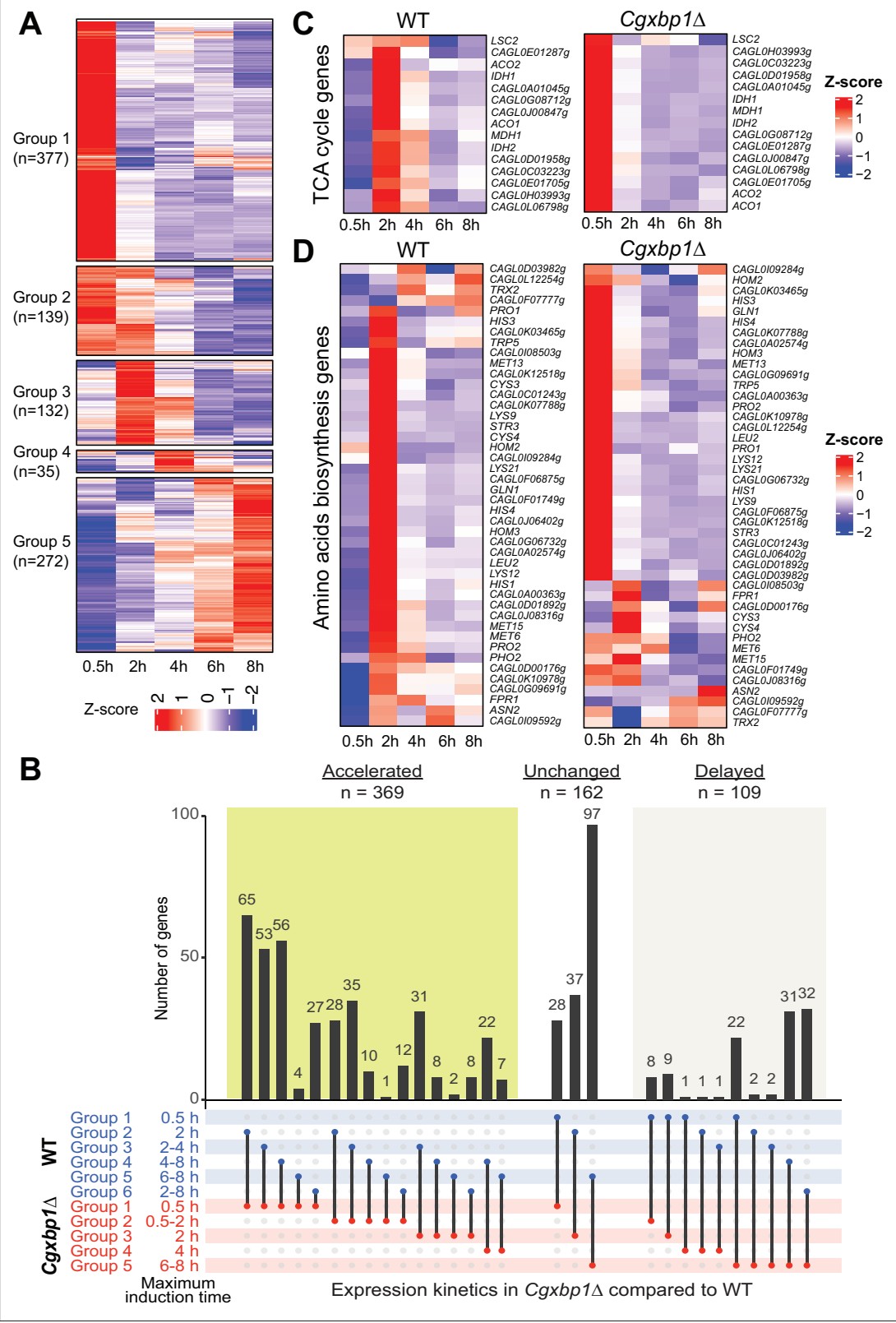

**Figure 3.** Loss of *CgXBP1* affects the expression level and timing of multiple genes of diverse physiological pathways upon macrophage phagocytosis. (**A**) A heatmap showing temporal expression patterns of transcribed genes in the *Cgxbp1Δ* mutant during 0.5–8 hr THP-1 macrophage infection in a time-course experiment. Groups were assigned by *k*-means clustering. (**B**) An UpSet plot showing the number of genes induced at the indicated time

*Figure 3 continued on next page*

*Figure 3 continued*

points in wild-type (WT) and the *Cgxbp1Δ* mutant during THP-1 macrophage infection. (**C and D**) Heat maps showing transcription activities of genes belonging to (**C**) tricarboxylic acid (TCA) and (**D**) amino acid biosynthesis during THP1 macrophage infection in wild-type and the *Cgxbp1Δ* mutant.

The online version of this article includes the following source data and figure supplement(s) for figure 3:

**Figure supplement 1.** CgXbp1 is essential for the chronological transcriptional response of *C. glabrata* during macrophage infection.

**Figure supplement 1—source data 1.** Table shown in *Figure 3—figure supplement 1B*.

Systematic GO analysis revealed multiple biological processes enriched among the genes with precocious activation in the *Cgxbp1Δ* mutant (0–0.5 hr). They include processes like energy generation, chromatin assembly, cellular respiration, and metabolism pathways such as the TCA cycle, acetate catabolism, amino acid, carboxylic acid, nucleotide, and trehalose biosynthesis (*Figure 3—figure supplement 1B*, *Supplementary file 11*). On the other hand, cell adhesion, host response, and biofilm formation genes, which were up-regulated in wild-type cells during the late infection stage did not show differential expression in the *Cgxbp1Δ* mutant within the 8 hr infection duration examined (*Supplementary files 2 and 11*).

Given that remodelling of carbon and nitrogen metabolism is crucial for the survival of fungal pathogens inside phagocytic cells (*Lorenz and Fink, 2001*; *Rai et al., 2012*; *Rubin-Bejerano et al., 2003*; *Seider et al., 2014*), we closely examined the expression patterns of the TCA cycle and amino acid biosynthesis genes in wild-type and the *Cgxbp1Δ* mutant during macrophage infection. In wild-type cells, most genes of these two metabolic pathways were temporally induced with the maximal induction at 2 hr (*Figure 3C and D*). By contrast, the induction of these genes was advanced to 0.5 hr (*Figure 3C and D*), and their overall expression levels were significantly higher (1.5–14.6 folds) in the mutant compared to wild-type. It is noteworthy that none of these TCA and amino acid biosynthesis genes (except for two genes - *GLN1* and *CAGL0D00176g*) were the direct binding targets of CgXbp1, while Xbp1 binds to the promoter of the transcription factor genes, whose *S. cerevisiae* orthologue regulates carbon metabolism (Mig1, Adr1, Rgm1 and Tye7) and amino acid biosynthesis (Leu3). Therefore, these results indicate that CgXbp1 negatively regulates the expression of TCA and amino acid

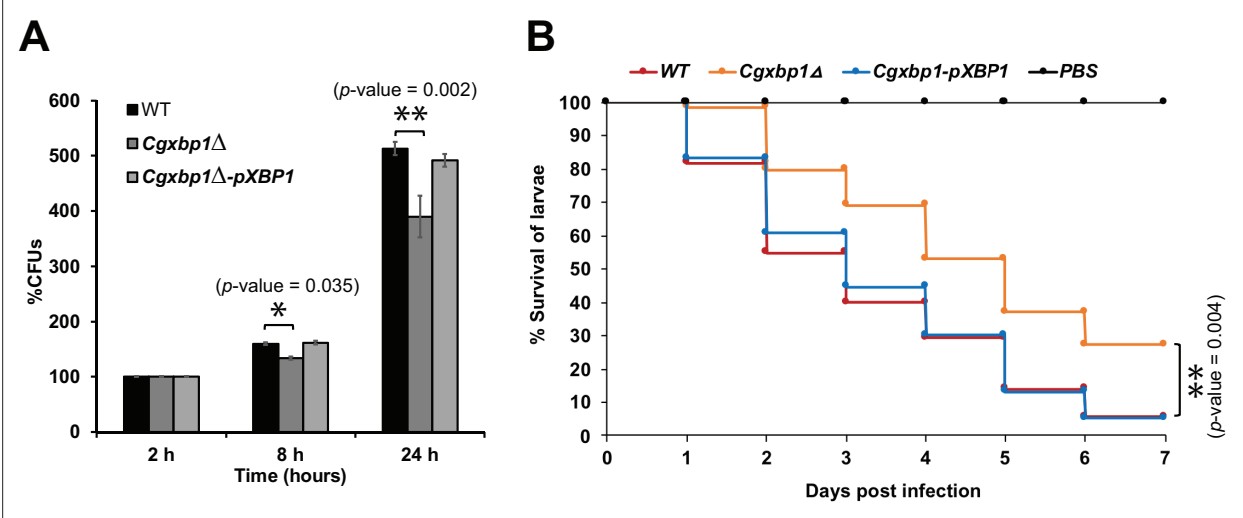

**Figure 4.** Loss of *CgXBP1* affects *C. glabrata* proliferation in human macrophages and attenuates virulence in the *Galleria mellonella* model of candidiasis. (**A**) Bar chart of colony forming units (CFUs) obtained from *C. glabrata* cells harvested from THP-1 macrophages at indicated time points. Error bars represent the standard error of the mean (± SEM) from three independent experiments. Statistical significance was determined by two-sided unpaired Student's *t*-test, p-value ≤0.05, p-value ≤0.01. (**B**) Cumulative survival curve of *G. mellonella* larvae infected with indicated *C. glabrata* strains. At least 16 larvae were used in each of the three independent infection experiments. The graph represents the percent survival of larvae infected with the indicated strains from three independent infection experiments. Statistical significance was determined by a two-sided unpaired Student's *t*-test, p-value ≤0.01.

The online version of this article includes the following figure supplement(s) for figure 4:

**Figure supplement 1.** Phagocytosis rate is not affected by CgXbp1 deletion.

biosynthesis genes indirectly through the transcription factors upon macrophage infection. Overall, the above results demonstrate that CgXbp1 is critical for the chronological transcriptional response of *C. glabrata* during macrophage infection.

## Loss of *CgXBP1* diminishes *C. glabrata* proliferation in human macrophages and attenuates virulence in the *Galleria mellonella* model of candidiasis

To examine if the altered transcriptional response in the *Cgxbp1Δ* mutant affects the survival of *C. glabrata* cells in macrophages, we compared the ability of wild-type and the *Cgxbp1Δ* mutant to survive in THP-1 macrophages. PMA-differentiated THP-1 macrophages were infected by wild-type and *Cgxbp1Δ* cells, and colony forming unit (CFU) assay was performed to determine the number of viable phagocytosed *C. glabrata* cells at 2-, 8-, and 24 hr post macrophage infection. No significant difference in CFUs between wild-type and *Cgxbp1Δ* cells was observed at 2 hr (*Figure 4—figure supplement 1A*), suggesting similar phagocytosis efficiency of THP-1 macrophages for the two strains. At 8 and 24 hr post-infection, wild-type cells exhibited ~1.6 and 5.1-fold increase in CFUs compared to that at 2 hr. Although the *Cgxbp1Δ* mutant was able to proliferate inside macrophages, it displayed significantly lower CFUs (~20%) at both time points (1.3 and 3.9-fold) (*Figure 4A*). The reductions were rescued in the *Cgxbp1Δ-pXBP1* complemented strain (*Figure 4A*). These results indicate that CgXbp1 is important for *C. glabrata* proliferation within macrophages.

We next examined the virulence of the wild-type and *Cgxbp1Δ* strains using the *Galleria mellonella* model of *Candida* infection (*Jacobsen, 2014*). We infected *G. mellonella* larvae with the wild-type, *Cgxbp1Δ*, and complemented strains, and monitored the morbidity and mortality of infected larvae over seven days. Although worms injected with wild-type or *Cgxbp1Δ C. glabrata* cells (but not phosphate buffered saline [PBS]) turned dark gray within 4–6 hr of infection due to melanin formation, which is a moth response to *C. glabrata* infection, and eventually died (*Figure 4B*), larvae injected with *Cgxbp1Δ* cells have a consistently slower mortality rate by ~20–30% compared to larvae infected by wild-type cells (*Figure 4B*), suggesting that the loss of Xbp1 function attenuated the virulence. The attenuated virulence was rescued in the complemented strain (*Figure 4B*). Therefore, CgXbp1 is important for the survival of *C. glabrata* in human macrophages and virulence in the in vivo infection model.

## CgXbp1 affects fluconazole resistance through repressing drug transporters' expression

We found that several genes associated with fluconazole resistance are CgXbp1 direct targets and/or have their transcription profiles altered in the *Cgxbp1Δ* mutant during macrophage infection (e.g. *CgAZR1*, *CgTPO1*, *CgFLR2*, *CgQDR2*, *CgPDH1*, *CgPDR13*, *CgERG6*, and *CgERG11* *Costa et al., 2016*; *Costa et al., 2013*; *Hallstrom et al., 1998*; *Miyazaki et al., 1998*; *Pais et al., 2016a*; *Figure 5A*, *Supplementary file 10*). Therefore, we examined whether CgXbp1 affects the resistance of *C. glabrata* to the antifungal fluconazole. Serial dilution spotting (*Figure 5B*) and MIC assays (*Figure 5C and D*) showed that the *Cgxbp1Δ* mutant had higher resistance to fluconazole compared to wild-type. Importantly, the altered resistance is not due to an intrinsic difference in growth rate between the two strains (as demonstrated by their indistinguishable growth rates in the absence of drug in liquid media in *Figure 5E*) or a general reduction in fitness under stressful conditions (as shown by plate test results in *Figure 4—figure supplement 1B*). However, in the presence of fluconazole, the *Cgxbp1Δ* mutant was able to initiate growth sooner than the wild-type (*Figure 5E*). Moreover, the *Cgxbp1Δ* mutant has a similar growth profile at the fluconazole concentrations of 24 and 32 µg/mL, whereas wild-type is more inhibited by 32 µg/mL than 24 µg/mL, suggesting that the *Cgxbp1Δ* mutant can better adapt and tolerate fluconazole than wild-type cells.

It is also interesting to note that there seems to be relatively more resistant colonies in the *Cgxbp1Δ* mutant as compared to wild-type in the spotting assay (*Figure 5B*). To confirm this, we performed a CFU assay by plating an equal number of exponentially growing wild-type, *Cgxbp1Δ* mutant, and complemented cells on YPD medium with or without fluconazole (64 µg/mL). The *Cgxbp1Δ* mutant displayed ~eightfold higher CFUs on fluconazole compared to that of wild-type and the complemented strain (*Figure 5F*), demonstrating the loss of CgXbp1 function led to a larger population of resistant cells. To better understand the molecular mechanism of CgXbp1-mediated fluconazole

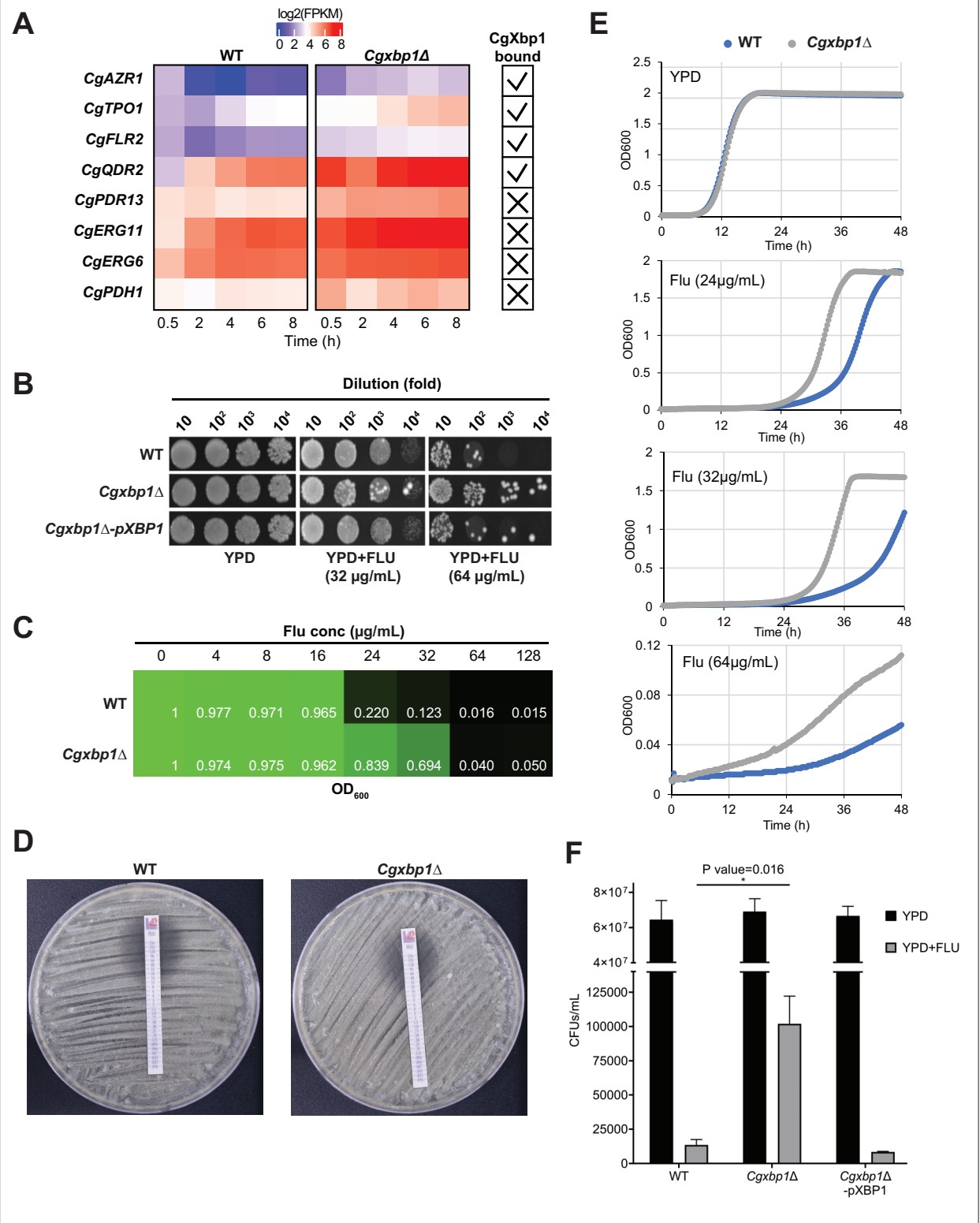

**Figure 5.** CgXbp1 regulates fluconazole resistance in *C. glabrata*. (**A**) Heatmap showing the genes related to fluconazole resistance with misregulated expression pattern in *Cgxbp1Δ* mutant. The ticks and crosses inside the boxes are displaying whether the gene is bound by CgXbp1. (**B**) Serial dilution spotting assay on YPD medium in the presence of different fluconazole concentrations (0, 32, or 64 μg/mL). (**C**) MIC$_{50}$ assay displaying growth of wild-type and *Cgxbp1Δ* mutant strains at indicated fluconazole concentration. (**D**) MIC$_{50}$ determination using fluconazole strips for wild-type and *Cgxbp1Δ* mutant strains. (**E**) Growth curve of wild-type and *Cgxbp1Δ* mutant in YPD medium in the presence or absence of fluconazole (24, 32, 64 μg/mL). (**F**)

*Figure 5 continued on next page*

*Figure 5 continued*

Bar graph displaying CFUs/mL obtained for indicated strains on YPD agar plates in the presence or absence of fluconazole (64 µg/mL) post 3 days of spread plating. Error bars represent mean ± SEM from three independent experiments. Statistical significance was determined by a two-sided unpaired Student's *t*-test, p-value ≤0.05.

resistance, RNAseq was performed on wild-type and *Cgxbp1Δ* cells grown in the presence or absence of fluconazole (64 µg/mL). The number of DEGs in wild-type and *Cgxbp1Δ* cells in response to fluconazole (1,163 and 1,246, respectively; *Figure 6A*, *Supplementary file 12*) and their enriched GO pathways are also similar between the two strains (*Figure 6B*, *Supplementary file 12*). The expression levels of fluconazole-responsive genes, including many ergosterol biosynthesis genes whose expression can influence fluconazole resistance, are also similar in the two strains (*Figure 6—figure supplements 1A and 2A*), indicating that the response to fluconazole of the *Cgxbp1Δ* mutant was not affected and that the resistance is not due to changes in the ergosterol level. Unexpectedly, there was no significant difference in the transcriptomes of the two strains in the presence of fluconazole (*Figure 6C*, *Figure 6—figure supplement 1B*), despite the fact that CgXbp1 expression was significantly induced (*Figure 6D and E*).

On the other hand, 135 genes were differentially expressed in the *Cgxbp1Δ* mutant during normal exponential growth (i.e. no fluconazole treatment) (*Figure 6C*) with up-regulated genes highly enriched with the 'transmembrane transport' function and down-regulated genes associated with different metabolic processes (e.g. carbohydrate, glycogen and trehalose) (e.g. carbon metabolism, nucleotide metabolism, and transmembrane transport, etc.) (*Supplementary file 13*). Interestingly, the TCA cycle and amino acid biosynthesis genes, whose expressions were accelerated in the *Cgxbp1Δ* mutant during macrophage (*Figure 3C and D*), were not affected by the loss of CgXbp1 function under normal growth conditions (i.e. in YPD media without fluconazole *Figure 6—figure supplement 2B*, *Supplementary file 12*), suggesting that the overall (direct and indirect) effects of CgXbp1 are condition-specific.

Of note, several multidrug transporter genes (*CgFLR1*, *CgTPO1,* and *CAGL0B02343g*), which have been associated with azole resistance (*Pais et al., 2016b*; *Vermitsky et al., 2006*), were significantly up-regulated in *Cgxbp1Δ* cells in the absence of fluconazole (*Figure 6C and F*), indicating that the *Cgxbp1Δ* cells have elevated drug efflux potentials. This is consistent with the better adaptability and tolerance of the *Cgxbp1Δ* mutant observed from the growth assay (*Figure 5E*). On the other hand, the expression of these efflux genes was not affected by CgXbp1 in the presence of fluconazole (*Figure 6F*). Therefore, the fluconazole resistance of *Cgxbp1Δ* is a result of elevated drug efflux. Taken together, the above results show that CgXbp1 not only orchestrates the temporal transcription response during macrophage infection but also governs fluconazole resistance in *C. glabrata* by repressing the expression of drug transport genes.

## Discussion

*C. glabrata* is well known for its ability to survive and grow inside phagocytic immune cells and to withstand azole antifungal drugs (*Kaur et al., 2005*; *Rai et al., 2012*). This work provides insights into the physiological events occurring at different stages of macrophage infection and identifies a novel transcription regulator important for responding to and proliferating within macrophages as well as the regulation of fluconazole resistance.

Through mapping genome-wide RNAPII occupancy, our result reveals that about 30% of *C. glabrata* genes are transcribed during the adaptation, survival, and growth inside the alien macrophage microenvironments. The number of genes may be an underestimate of the overall response, as the RNAPII ChIP-seq method has a narrow, low detection range relative to RNAseq and may not be able to detect lowly transcribed genes. Nevertheless, our data reveal dynamic temporal responses during macrophage infection. At the most immediate response (0–0.5 hr), *C. glabrata* activates adherence-related genes to initiate adhesion to host surfaces. Concurrently, *C. glabrata* elicits specific responses to the nutrient-limiting microenvironment inside macrophages. This immediate response is followed by *C. glabrata* efforts to deal with oxidative and DNA-damage stresses (0–2 hr), and at this time the phagocytosed *C. glabrata* cells are arrested at the G1-S phase of the cell cycle. Subsequently (2–4 hr), *C. glabrata* undergoes transcriptional remodeling to adjust its carbon metabolism, presumably to

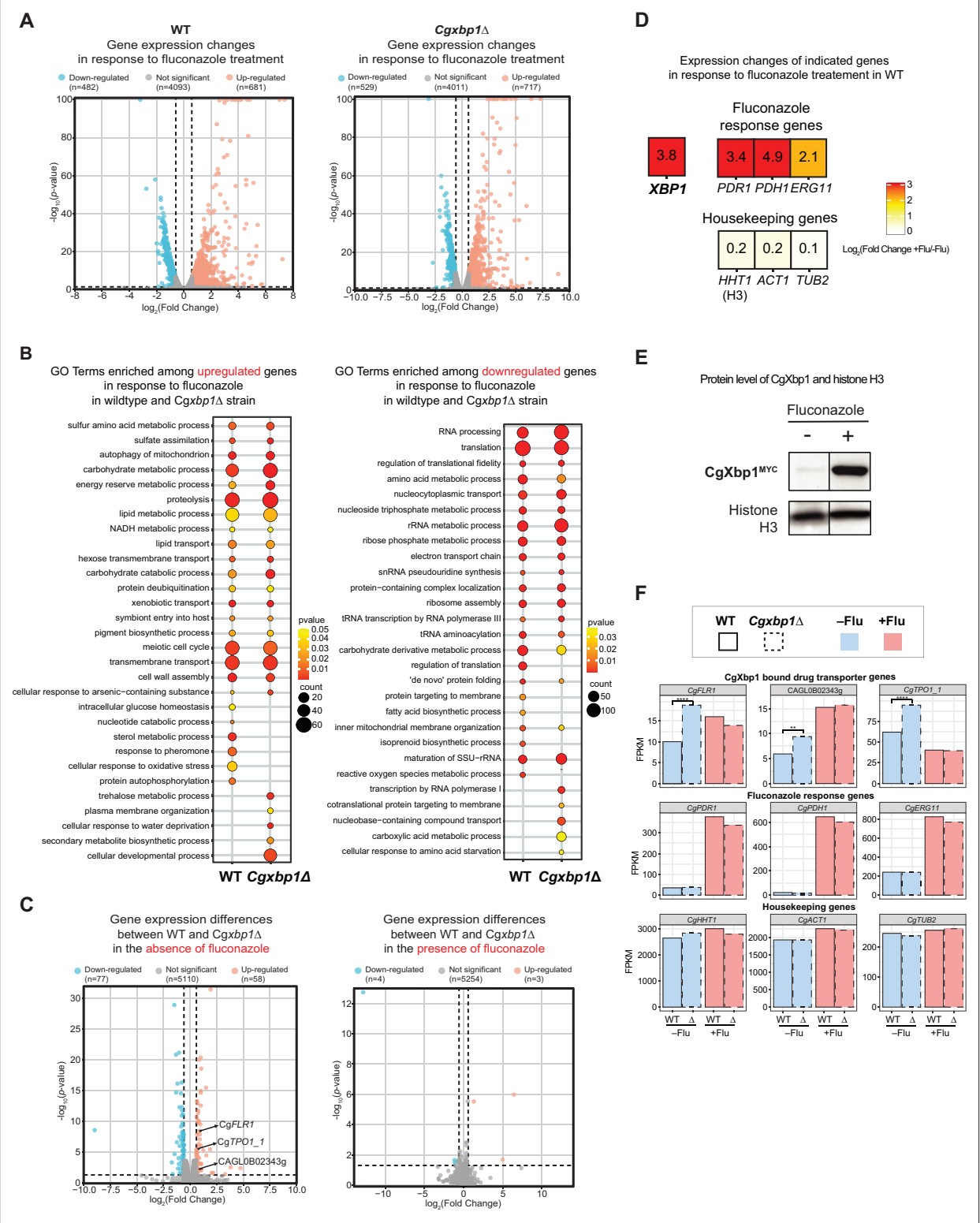

**Figure 6.** RNAseq analysis revealed up-regulation of drug transporters in *Cgxbp1Δ*. (**A**) Volcano plots showing expression changes of all genes and their p-values in wild-type and *Cgxbp1Δ* strains grown under conditions with and without fluconazole. Down- and up-regulated genes are coloured blue and red, respectively, while genes with no significant change in expression are in gray. The number of genes in each group is indicated in parentheses. (**B**) GO terms enriched among the up- (left panel) and down-regulated (right panel) in response to fluconazole treatment in wild-type and *Cgxbp1Δ* strains. The colour scale depicts p-values and the size of the circles shows the number of DEGs associated with each gene ontology (GO) term. (**C**) Volcano plots showing expression changes of all genes and their p-values comparing between wild-type and *Cgxbp1Δ* strains grown in the absence (left

*Figure 6 continued on next page*

*Figure 6 continued*

panel) and presence (right panel) of fluconazole. Down- and up-regulated genes are coloured blue and red, respectively, while genes with no significant change in expression are in gray. The number of genes in each group is indicated in parentheses. (**D**) Heat boxes showing the expression changes of *CgXBP1* (*XBP1*), fluconazole response genes [*CgPDR1* (*PDR1*), *CgPDH1* (*PDH1*), *CgERG11* (*ERG11*)], and housekeeping genes [*CgHHT1* (*HHT1*), *CgACT1* (*ACT1*), *CgTUB2* (*TUB2*)] after fluconazole treatment. The levels of change are expressed in $\log_2$ fold change and presented in a coloured scale and within the box. (**E**) Western Blot showing the expression level of the CgXbp1$^{MYC}$ and histone H3 proteins upon fluconazole treatment. (**F**) Bar chart displaying the gene expression of drug transporter genes (first row), fluconazole response genes (second row), and housekeeping genes (last row) for wild-type (WT) and Cg*xbp1Δ* (Δ) mutant in the presence or absence of fluconazole. Wild-type and Cg*xbp1Δ* are framed with solid line borders and dashed line borders, respectively. No fluconazole treatment (-flu) is coloured blue and fluconazole treatment (+flu) is coloured red. Statistical significance was calculated by *p*-value in DEseq2, **<i>p-value ≤0.01, ****<i>p-value ≤0.0001.

The online version of this article includes the following source data and figure supplement(s) for figure 6:

**Source data 1.** Original files for the western blots shown in *Figure 6E*.

**Source data 2.** A Microsoft Word file containing original western blots for *Figure 6E*, indicating the relevant bands and treatments.

**Figure supplement 1.** *Cgxbp1* deletion does not affect the overall fluconazole response.

**Figure supplement 2.** Effect of *Cgxbp1Δ* under normal growth conditions.

generate energy for future challenges, growth, and/or proliferation. Consistently, processes necessary for growth such as global transcription, ribosome biogenesis, and copper and iron ion homeostasis are activated around this time (2–6 hr). In particular, iron homeostasis probably has an overwhelming role in the pathogenic adaptation of *C. glabrata*, as a large number of iron-responsive genes (n=154) including those with iron uptake functions are transcribed during macrophage infection (*Figure 1—figure supplement 3*, *Supplementary file 3*). Iron bioavailability in macrophages is presumably limited, and high-affinity iron uptake is pivotal for virulence in *C. glabrata* (*Bairwa et al., 2017*; *Srivastava et al., 2014*). Therefore, these transcriptional activities indicate that *C. glabrata* is still adapting to the macrophage microenvironment at this stage of infection. Lastly (8 hr), *C. glabrata* induced genes associated with cell proliferation and biofilm formation, implying that they have overcome macrophage attacks and are ready to grow and divide.

It is noteworthy that *C. glabrata*'s transcriptional responses to macrophages are concerted at stages. For example, despite sensing starvation within the first 0.5 hr upon macrophage engulfment, *C. glabrata* does not activate alternate carbon catabolic pathways until 2 hr. In addition, gene expression and translation-related genes show the lowest transcription levels (i.e. RNAPII occupancy) at this immediate stage (0.5 hr) relative to the other time points (Group 6 genes in *Figure 1C and D*), indicating global suppression of gene expression in *C. glabrata* upon macrophage phagocytosis. A recent study showed that the fungal pathogen *Cryptococcus neoformans* also down-regulates translation during exposure to oxidative stress (*Leipheimer et al., 2019*). The global suppression of gene expression under stressful conditions probably helps pathogens to reserve energy and resources for coping with stress such as the hostile, nutrient-limiting macrophage environment.

Transcriptional responses are determined by the overall TFs activities in a cell. The RNAPII profiles revealed a panel of 53 TF genes expressed during macrophage infection. In particular, 39 of them were temporally induced (*Figure 2A*) and are promising candidate regulators for the stage-specific transcription responses. The orthologues of several transcription factors in different fungal pathogens are involved in stress response, macrophage killing, and virulence. For example, *C. glabrata* Yap6 (*Zhou et al., 2020*) and the conserved CgRim101 orthologues in several fungal pathogens control pH response and is important for host adaptation and virulence (*Davis et al., 2000*; *O'Meara et al., 2014*; *Peñalva et al., 2008*; *Yuan et al., 2010*); *C. albicans* Rtg1 is necessary for adaption to ROS and, therefore, host colonisation and virulence *Oneissi et al., 2023*; *Pérez et al., 2013*; *Pérez and Johnson, 2013*; the Zap1 orthologue of *Cryptococcus gattii* controls zinc homeostasis and affect virulence; *C. albicans* Tye7 activates glycolysis genes during macrophage infection to induce glucose competition and, consequently, macrophage killing (*Tucey et al., 2018*) and is required for full virulence (*Askew et al., 2009*). It is interesting to note that the glycolysis and gluconeogenesis pathways were both induced during infection but the maximal induction occurred at different times of the early to intermediate infection stage, suggesting that that phagocytosed *C. glabrata* is striving to adapt to the nutrient available within the macrophage micro-environment and maintain metabolic homeostasis. A similar observation was found for *C. albicans* which undergoes multiple rounds of metabolic reprogramming during macrophage infection (*Tucey et al., 2018*).

Moreover, transient fluctuation in peroxisome number that indicates metabolic switching has also been reported in phagocytosed *C. glabrata* (*Roetzer et al., 2010*). In *C. albicans*, a recent study demonstrated glucose competition as a key strategy of phagocytosed cells to kill macrophages (*Tucey et al., 2018*). The up-regulation of the glycolysis genes in our infection time course experiment (*Figure 1—figure supplement 4A*) suggests that *C. glabrata* also metabolises glucose within macrophages. However, unlike *C. albicans*, *C. glabrata* does not cause massive macrophage killing but instead proliferates inside macrophages (*Kaur et al., 2007*; *Seider et al., 2011*). It seems that *C. glabrata* can acquire its carbon nutritional needs without causing glucose depletion in the host, and this may be related to its dependency on autophagy to survive and proliferate in macrophages (*Roetzer et al., 2010*).

Of note, four TFs (CgHap3, CgXbp1, CgDal80, and CgRpa12) were strongly activated at the earliest infection stage. In *S. cerevisiae*, the orthologues of three TFs play negative regulatory roles (i.e. transcriptional repressors *Mai and Breeden, 1997*; *Marzluf, 1997*; *Yadav et al., 2016*), suggesting that transcriptional repression plays crucial roles in shaping the early and, consequently, overall transcriptional response to macrophages (*Figure 7A*). In addition, the *C. albicans* HAP3 gene has been implicated in modulating immune cell recognition and response during phagocytosis by dendritic cells through controlling cell wall remodelling genes (*Tierney et al., 2012*). The upregulation of CgHap3 in *C. glabrata* during the early stage of macrophage infection suggests a similar conserved strategy to escape immune cell recognition.

The importance of transcriptional repression was confirmed through the functional characterization of CgXbp1, which revealed that many genes were precociously activated in the *Cgxbp1Δ* mutant during macrophage infection. In addition, our ChIP-seq results showed that CgXbp1 directly binds to the promoter of many TFs, indicating that CgXbp1 indirectly represses the activation of many gene regulatory networks. Therefore, CgXbp1 exerts twofold regulation: directly controlling downstream effector genes and indirectly affecting gene networks of diverse pathways *via* their hierarchies (i.e. TFs).

It is noteworthy that the *S. cerevisiae* orthologues of the other two uncharacterized proteins, CgDal80 and CgRpa12, negatively regulate genes involved in nitrogen and lipid metabolism, respectively (*Hofman-Bang, 1999*; *Yadav et al., 2016*). We postulate that these repressors also act in the same fashion as CgXbp1 to delay the induction of other groups of genes (such as nitrogen and lipid metabolism genes) that are not immediately necessary for the immediate response but are required at later stages of macrophage infection and/or for proliferation. Therefore, our overall results suggest a mechanistic model for *C. glabrata*'s macrophage response, in which global transcriptional repression is established at the early infection stage to withhold activation of genes whose functions are not immediately needed (*Figure 7A*). The repression is subsequently relieved by transcriptional down-regulation and protein turnover of the repressors and/or through enhanced activation by pathway-specific transcriptional activators. The interplay between transcriptional activators and repressors (*Figure 7A*) is crucial in shaping the dynamic transcriptional response during macrophage infection.

Our results further highlight the physiological significance of the temporal transcriptional responses of *C. glabrata* during macrophage infection. The precocious activation of numerous virulence genes, such as those involved in carbon and nitrogen metabolism, in the *Cgxbp1Δ* mutant resulted in decreased proliferation within phagocytic cells. This suggests that timely and coordinated expression of virulence genes is crucial for *C. glabrata*'s survival and pathogenic response during macrophage infection. Presumably, the pathogen needs to strategize the utilization of cellular resources to survive and counteract host attacks during infection, and this may also be the reason for the reduced virulence in the Galleria infection model.

In addition to its role in controlling macrophage response, CgXbp1 also plays a significant role in drug resistance. Our findings demonstrate that CgXbp1 does not affect the expression of ergosterol biosynthesis genes but primarily exerts its effect by repressing the expression of drug transporters, thereby controlling the efflux potential of the cell (*Figure 7B*). Interestingly, even though the levels of CgXbp1 were significantly increased in the presence of fluconazole, we observed no changes in the global transcriptome of the *Cgxbp1Δ* mutant compared to the wild type. This suggests that fluconazole fully suppresses CgXbp1's transcriptional activity (*Figure 7B*). Presumably, the heightened level of CgXbp1 facilitates rapid gene repression when fluconazole levels decrease.

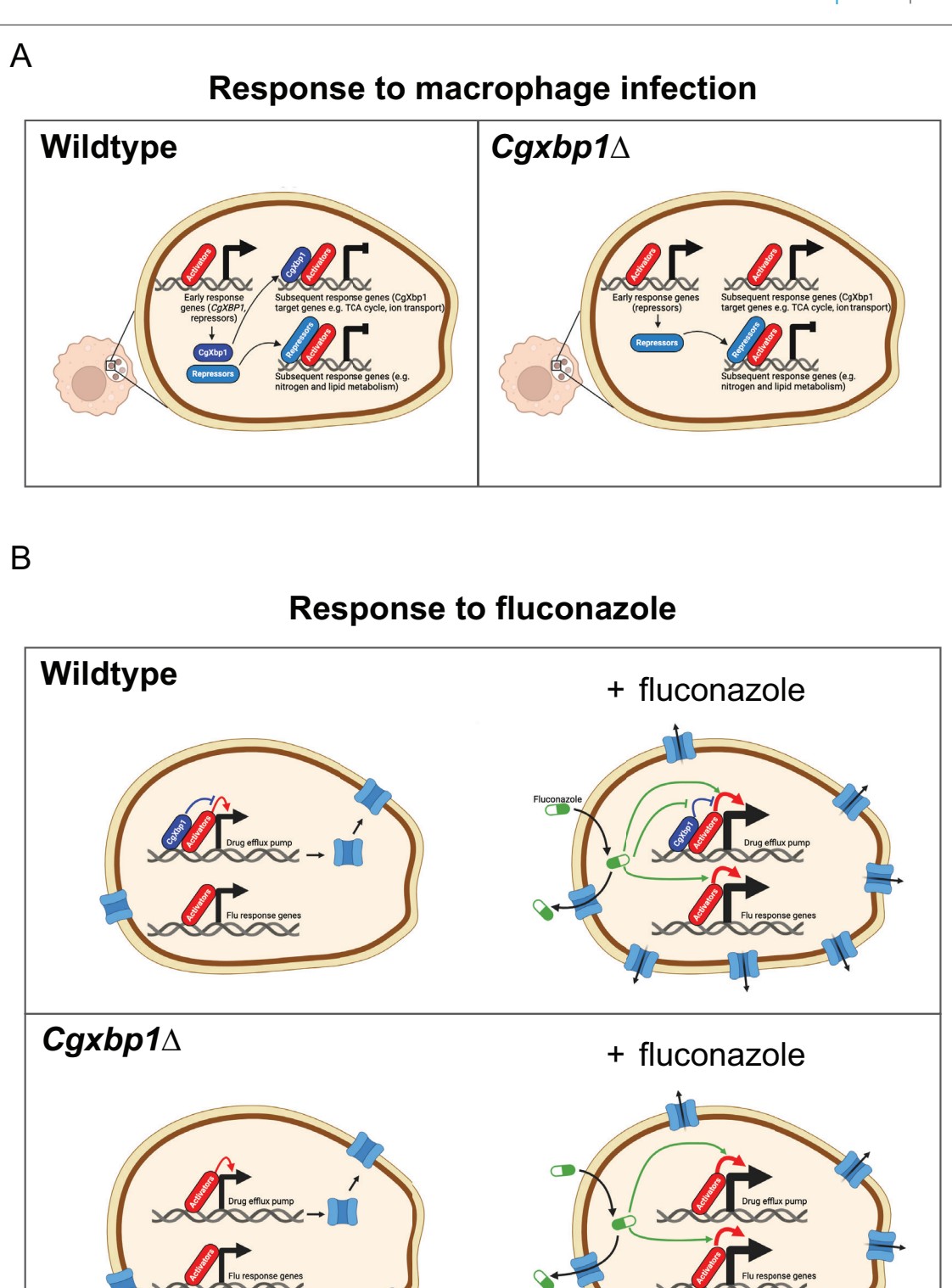

**Figure 7.** Schematic models for the role of CgXbp1 during macrophage infection and in fluconazole resistance. (**A**) Upon macrophage infection, the gene of *Cgxbp1* and several transcriptional repressors are transcriptionally induced at the early stage of macrophage infection. CgXbp1 and the other repressors then inhibit the expression of their target genes, which are expressed at the subsequent stages. In the absence of CgXbp1, induction of these intermediate/late response genes is temporally advanced. Therefore, CgXbp1 (presumably the other repressors) acts to delay the expression of

*Figure 7 continued on next page*

*Figure 7 continued*

different subsets of genes until later. When *C. glabrata* cells are adjusted to the macrophage host, CgXbp1 repression of its target genes is relieved through transcriptional down-regulation, allowing their functions to be expressed. We propose that, in addition to activators, repressors also play an important role in orchestrating the dynamic temporal transcription response of *C. glabrata* during macrophage infection. (**B**) CgXbp1 also has a role in fluconazole resistance. In the absence of fluconazole, CgXbp1 negatively regulates the expression of several drug efflux transporters, which are expressed at low but detectable levels in the wild-type. On the other hand, *Cgxbp1Δ* expresses the drug transporters at a higher level than wild-type cells, so the *Cgxbp1Δ* cell has more efflux transporters and, therefore, can pump out fluconazole (and other drugs) more rapidly. Consequently, *Cgxbp1Δ* cells can adapt better and initiate growth faster than wild type when exposed to fluconazole. After a prolonged exposure to fluconazole, our RNAseq data revealed that both wild-type and *Cgxbp1Δ* cells produce the same transcriptional response to fluconazole, indicating that CgXbp1 is not involved in the response (flu-responsive genes). The lack of CgXbp1 effect also suggests that CgXbp1's function is inhibited by fluconazole. We propose that CgXbp1 controls the drug efflux potential in wild-type *C. glabrata* cells. This model was created in BioRender (https://biorender.com/y60k027 and https://www.biorender.com/q03d823).

Collectively, this study unveils a multifaceted transcriptional regulator that is crucial for survival within macrophages and conferring antifungal drug resistance, two key properties unique to the human fungal pathogen *C. glabrata*.

## Methods
### Culture conditions for *C. glabrata* and THP-1 macrophages

*C. glabrata* strain BG2 was used as the wild-type in all experiments and the parental strain for genetic modifications. The strains generated and used in this study can be found in *Table 3*. A single colony of indicated *C. glabrata* strains was cultured overnight (14–16 hr) in YPD medium at 30 °C and 200 rpm in a shaker incubator. The THP-1 cell line was obtained from ATCC (TIB 202). THP-1 cells were grown in RPMI medium supplemented with 20 mM glutamine, antibiotic (penicillin-streptomycin, 1 X), and 10% heat-denatured serum at 37 °C with 5% $CO_2$ in a cell culture incubator.

### *C. glabrata* infection of macrophages for RNAPII ChIP-seq and CgXbp1 ChIP-seq

Macrophage infection assays were done as described previously (*Rai et al., 2013*). THP-1 monocytes were grown till 80% confluence, harvested, and re-suspended in RPMI medium at a cell density of $1×10^6$ cells/mL. For macrophage differentiation, Phorbol-12-myristate 13-acetate (PMA) was added to the THP-1 monocytes at a final concentration of 16 nM. Approximately 10 million cells were seeded in 100 mm culture dishes and incubated for 12 hr at 37 °C with 5% $CO_2$ in a cell culture incubator. Subsequently, the culture medium was replaced with a fresh pre-warmed complete RPMI medium to remove PMA, and cells were allowed to recover in the absence of PMA for 12 hr. Macrophage differentiation and adherence were confirmed under the microscope. Overnight grown *C. glabrata* cells were harvested, washed with PBS and finally suspended in complete RPMI medium at a density of $10^8$ yeasts/ml. To infect THP-1 macrophages, 500 μL yeast cell suspension ($5×10^7$ yeast cells) was added to each culture dish containing differentiated THP-1 macrophages at a MOI of 5:1. Post 0.5 hr macrophage infection (referred to as 0.5 hr), THP-1 macrophages were crosslinked using formaldehyde at a final concentration of 1% for 20 min before 1.5 mL of 2.5 M glycine (a final concentration of 320 nM) was added to stop the crosslinking reaction. For the remaining time points (2 hr, 4 hr, 6 hr, and 8 hr), culture dishes were washed gently with PBS three times to remove non-phagocytosed yeast cells, and the medium was replaced with fresh pre-warmed RPMI medium. The infected culture was further incubated until the indicated infection times before formaldehyde crosslinking as described

**Table 3.** Strains used in this study.

| Strain Number | Strain Name | Genotype |
|---|---|---|
| CWF28 | wild-type | BG2 |
| CWF236 | *Cgxbp1Δ* | *CgXBP1*::hph1 |
| CWF1325 | *CgXBP1*<sup>MYC</sup> | *CgXBP1*<sup>MYC</sup>; hph1 |
| CWF1327 | *Cgxbp1Δ*-p*XBP1* | *CgXBP1*::hph1, pCN-*CgXBP1* |

above, infected macrophage cultures were harvested and washed three times with ice-cold TBS before storing in –80 °C freezer till chromatin extraction. To prepare *C. glabrata* infection samples for CgXbp1 ChIP-seq, 20 million cells were seeded in 100 mm Petri dishes. PMA induction procedure was the same as described above for RNAP II ChIP-seq. About 100 million yeast cells were used to infect differentiated THP-1 macrophages and incubated for 2 hr. After the formaldehyde crosslinking and PBS wash, the infected macrophages were harvested by a scrape, washed by PBS, and stored in the –80 °C freezer.

## ChIP and Illumina sequencing library preparation

Chromatin was prepared using a previously described protocol (*Fan et al., 2008*) with modifications. Briefly, the infected macrophage cell pellet was resuspended in 400 µL FA lysis buffer and 10 µL of 100 mM PMSF solution in the presence of 100 µL equivalent zirconium beads and lysed using six 3 min cycles at maximum speed in a Bullet Blender (Next Advance) homogeniser with at least 3 min of cooling on ice in between each cycle. Cell lysate was transferred to a new 1.5 mL tube and centrifuged at 2500 g for 5 min in a microcentrifuge. The supernatant was discarded, and the resultant pellet was re-suspended in 500 µL FA lysis buffer, and then transferred to a 2 mL screw-cap tube. Sonication was carried out to shear the crosslinked chromatin (cycles of 10 s on and 15 s off sonication for a total of 30 min sonication time), and chromatin was stored in the –80 °C freezer until use. Chromatin immuno-precipitation was carried out using 2 µL of a commercially available anti-RNA polymerase II subunit B1 phospho-CTD Ser-5 antibody (Millipore, clone 3E8, cat. no. 04–1572) for RNAP II, and anti-MYC tag antibody (Santa Cruz, cat. no. 9E10) for CgXbp1[MYC]. The sample was gently mixed on an end-to-end rotator at room temperature for 1.5 hr, and 10 µL of packed protein A sepharose beads (GE Health-care cat. no. 17-0618-01) were then added. The mixture was further incubated at room temperature for another 1.5 hr with gentle mixing. Immuno-precipitated material was washed twice with FA lysis buffer (150 mM NaCl), and once with FA lysis buffer (500 mM NaCl), LiCl wash buffer, and TE buffer before elution in 100 µL of elution buffer, as described previously (*Wong and Struhl, 2011*). Eluted DNA was decrosslinked at 65 °C overnight and purified using the QIAGEN PCR purification kit (cat. no. 28104). Sequencing library was generated using a multiplex Illumina sequencing protocol (*Wong et al., 2013*) and sequenced using the Illumina HiSeq2500 platform at the Genomics and Single Cells Analysis Core facility at the University of Macau.

## Bioinformatics and ChIP-seq data analyses

Raw fastq sequences were quality-checked using FastQC (http://www.bioinformatics.babraham.ac.uk/projects/fastqc/) aligned against the *C. glabrata* reference genome (CBS138_s02-m07-r06) using bowtie2 (*Langmead and Salzberg, 2012*). To visualize the ChIP-seq data on the IGV (integrated genome viewer *Thorvaldsdóttir et al., 2013*), aligned reads were processed by MACS2 (*Zhang et al., 2008*), and BigWig files were generated using 'bedSort' and 'bedGraphToBigWig' commands from UCSC Kent utils (*Kent et al., 2010*). Samtools (version 1.9) was used to index the resultant BAM file and check for alignment statistics. For RNAPII ChIP-seq analysis, elongating RNAPII occupancy was measured by first counting the number of reads over the gene body for all annotated genes (n=5311) and then normalising to gene length and sequencing depth using an in-house Perl script (https://github.com/zqmiao-mzq/perl_tools/blob/master/zqWinSGR-v4.pl; *zqmiao-mzq, 2021*), and was expressed as normalised RNAPII ChIP-seq read counts. RNAPII ranked from high to low as shown in *Figure 1—figure supplement 1A*, and manually inspected on the IGV to empirically determine a filtering cut-off that can reliably identify genes with significant and true RNAPII ChIP-seq signals. The normalised RNAPII ChIP-seq read counts values ≥12 and ≥25 were determined for wild-type and the *Cgxbp1Δ* mutant, respectively. These values are approximately three times higher than the ChIP-seq signal at background regions (3.2 and 7.0 for wild-type and the *Cgxbp1Δ* mutant, respectively). To ensure that lowly expressed but transcriptionally induced genes were not missed, we searched for genes with the high standard deviation among RNAPII binding signals across the five time points and empirically determined a cut-off (SD ≥2.25 and ≥4.00 for wild-type and the *Cgxbp1Δ* mutant, respectively) that includes most, if not all, genes with significant active transcription and/or changes in its level across the time course. This standard deviation-based approach identified 68 and 38 additional genes for wild-type and the *Cgxbp1Δ* mutant, respectively. The lists of transcribed genes for wild-type and the *Cgxbp1Δ* strains are given in *Supplementary files 1 and 10*, respectively. Fold changes

for the time course experiment were calculated with respect to 0.5 hr, while folding changes for DEGs relative to WT (i.e. Δ/WT). For the z-score plots, only genes whose expression changes at least twofolds between any two or more time points during the macrophage infection experiment were included. Z-scores were generated using the row clustering option in FungiExpresZ (*Parsania et al., 2023*). Heatmap, k-means clustering, and correlation plots were generated using an online tool Fungi-ExpresZ (https://cparsania.shinyapps.io/FungiExpresZ/). GO-term enrichment, and GO slim mapping analyses were performed on the *Candida* genome database (*Skrzypek et al., 2017*; http://www.candidagenome.org) and FungiDB (*Stajich et al., 2012* ;https://fungidb.org/fungidb/). Transcription regulatory networks between transcription factors and their target genes (*Figure 2—figure supplement 1*) were generated by the 'Rank by TF' function of PathoYeastract (*Monteiro et al., 2020*; http://pathoyeastract.org/cglabrata/formrankbytf.php) using published regulatory information of *S. cerevisiae*. For CgXbp1 ChIP-seq analysis, peak calling was done by MACS2 (*Feng et al., 2012*) using the parameters [macs2 pileup --extsize 200] and then normalized [macs2 bdgopt]. ChIP signal intensity at the 200 bp flanking regions of the peak summit from both replicates was used to determine the correlation between the biological replicates (*Figure 1—figure supplement 6*).

MACS2 (*Feng et al., 2012*) was used with the –nomodel setting to identify CgXbp1 binding sites from ChIPseq data. The identified binding sites were evaluated manually on the genome browser. Binding sites that have a peak with an irregular peak shape, a very low signal-tobackground noise ratio, and a similar signal pattern to the input DNA control were removed. Peak calling was carried out on biological repeats and the output peak lists were compared between repeats. The peaks that are present in biological repeats were included in the subsequent analysis. Identification of target genes was carried out using an in-house script (https://github.com/RaimenChan/Xbp1_project; copy archived at *Chen, 2024*) by mapping peaks to the closest gene within 1 kb upstream of its translation start site (i.e. ATG). In cases when a peak is located at a divergent promoter of two genes and the distance from the peak to the ATG of both genes is within 1 kb, then both genes are included as the CgXbp1 target.

## Generation of the CgXbp1^MYC, *Cgxbp1Δ*, and *Cgxbp1Δ-pXBP1* complemented strains

CgXbp1^MYC strain was generated as described previously (*Qin et al., 2019*). Briefly, a transformation construct was generated using 1 kb of 5' and 3' fragments, flanking the stop codon of *CAGL0G02739g* gene, with a 'MYC-*hph*' cassette between the two fragments. The 5' fragment for CgXbp1^MYC strain was amplified using primers 5'-ATATCGAATTCCTGCAGCCCTCCATGGTACATTGCAAAAC-3', and 5'-TTAATTAACCCGGGGGATCCGCACATTCTCTTGAAGATGGG-3' from *CAGL0G02739g* gene, and 3' fragment was same as for *Cgxbp1Δ* mutant. Hygromycin-resistant yeast colonies were selected, and tagging was confirmed by PCR and Sanger sequencing. To create the *Cgxbp1Δ* mutant, 1 kb of 5' and 3' flanking regions of the *CAGL0G02739g* gene were amplified using PCR with the primers 5'-ATATCGAATTCCTGCAGCCCGGCCAACCCCACTTCGAGGA-3' and 5'-TTAATTAACCCGGGGGATCCGTTAGTGATTTTGTAGTATGG-3' for the 5' flanking region and 5'-GTTTAAACGAGCTCGAATTCTCAAACATAATATAGTCATC-3' and 5'-CTAGAACTAGTGGATCCCCCGAGAAGTTTTGGGTTGTACG-3' for the 3' flanking region. A transformation construct was created as described previously (*Qin et al., 2019*) using the '*hph*' cassette encoding hygromycin resistance as the selectable marker and used to transform the *C. glabrata* wild-type strain (BG2). Hygromycin-resistant yeast colonies were checked for deletion of the *CgXBP1* gene using PCR with the primers from gene internal regions (5'-TGGTGCTTTGGACGCTACAT-3' & 5'-TCATCGCAAAAGCAATTGGACA-3'). To generate the complemented strain, *CgXBP1* ORF was first amplified using the forward (5'- GAATTCATGAGACTCACAGACTCGCCGCT-3') and reverse (5'- GTCGACTTACACATTCTCTTGAAGATGGGT-3') primers from *C. glabrata* genomic DNA, digested with *EcoR*I and *Sal*I, and cloned between *EcoR*I and *Sal*I restriction sites of a CEN/ARS episomal plasmid, pCN-PDC1. The resultant plasmid, pXBP1, carrying *CgXBP1* ORF was transformed into *Cgxbp1Δ* mutant, and the resultant transformed complemented strain was selected on YPD plates carrying NAT (100 µg/mL).

## Cell cycle analysis of intracellular *C. glabrata* cells in THP-1 macrophages

For cell cycle analyses, THP-1 macrophages were infected with *C. glabrata* cells in a 24-well cell culture plate as described above. In control wells, we inoculated an equal number of *C. glabrata* cells to RPMI medium. Post 2 hr incubation, *C. glabrata* cells were harvested and washed twice with 1 mL PBS. Next, harvested cells were fixed by re-suspending them in 1 mL of 70% ethanol, followed by incubation at room temperature on a rotator for 60 min. Fixed cells were pelleted and re-suspended in 1 mL of 50 mM sodium citrate (pH 7.0), and were sonicated for 15 s at 30% power to re-suspend cell aggregates. Subsequently, samples were treated with an RNase cocktail (0.3 μL, Ambion cat. no. AM2286) at 37 °C for 1 hr to remove RNA, washed with PBS, and stained with propidium iodide (PI) for 1 hr. Cells were then passed through a 40 mm membrane filter and were analysed on the BD Accuri C6 flow cytometer (excitation: 488 nm Laser, filter: 585/40, and detector: FL2).

## *C. glabrata* infection of macrophages for determining viability using colony forming unit assay

Macrophage fungi infection assays were done as described earlier (*Rai et al., 2013*). To prepare macrophages for infection assay, THP-1 monocytes were grown till 80% confluence, harvested, and resuspended to a cell density of $10^6$ cells/ml in a complete RPMI medium. Phorbol-13-myrstyl-acetate (PMA) was added to the cell suspension to 16 nM final concentration, mixed well, and 1 million cells were seeded in each well of a 24-well cell culture plate. Cells were incubated for 12 hr in a cell culture incubator, the medium was replaced with fresh pre-warmed complete RPMI medium, and cells were allowed to recover from PMA stress for 12 hr. Macrophage differentiation and adherence were confirmed under the microscope. Overnight grown *C. glabrata* cells were harvested, washed with PBS, and adjusted to $2×10^6$ yeasts/ml by cell counting using a hemocytometer and resuspended in a complete RPMI medium. 100 μL yeast cell suspension was added to each well of the 24-well culture plate containing differentiated THP-1 macrophages. Post 2 hr co-incubation, wells were washed three times with PBS to remove non-phagocytosed yeast cells, and the medium was replaced. At the indicated time post-infection, the supernatant was aspirated out from the wells, and macrophages were lysed in sterile water and incubated for 5 min for lysing the macrophages. Lysates containing fungal cells were collected, diluted appropriately in PBS, and plated on YPD mediums. Plates were incubated for two days at 30 °C and colonies were counted after 48 hr. The viability of *C. glabrata* cells was determined by comparing the colony-forming units.

## *Galleria mellonella* infection assay for virulence analyses

Indicated *C. glabrata* strains were grown in YPD medium overnight, washed with PBS thrice, and resuspended in PBS to a final cell density of $10^8$ cells/ml. Next, 20 μL of this cell suspension carrying $2×10^6$ *C. glabrata* cells were used to infect *G. mellonella* larvae. The infection was carried out three independent times, each on 16–20 larvae. An equal volume of PBS was injected into the control set of larvae. Infected larvae were transferred to a 37 °C incubator, and monitored for melanin formation, morbidity, and mortality for the next seven days at every 24 hr. The number of live and dead larvae was noted for seven days, and the percentage of *G. mellonella* larvae survival was calculated.

## Serial dilution spotting assay

*C. glabrata* strains were grown in YPD medium for 14–16 hr at 30 °C under continuous shaking at 200 rpm. Cells were harvested from 1 ml culture, washed with PBS, and were diluted to an $OD_{600}$ of 1. Next, five 10-fold serial dilutions were prepared from an initial culture of 1 $OD_{600}$. Subsequently, 3 μL of each dilution was spotted on YPD plates with or without fluconazole (32 & 64 μg/mL). Plates were incubated at 30 °C and images were captured after 2–8 days of incubation.

## Growth curve analyses

A single colony of the indicated strains was inoculated to liquid YPD medium and grown for 14–16 hr. The overnight grown culture was used to inoculate to YPD medium with or without 64 μg/mL fluconazole at an initial $OD_{600}$ of 0.1 in a 96-well culture plate. The culture plate was transferred to a 96 well-plate reader, Cytation3, set at 30 °C and 100 rpm. The absorbance of cultures was recorded at

$OD_{600}$ nm at regular intervals of 30 min over a period of 48 hr. Absorbance values were used to plot the growth curve.

## Minimum inhibitory concentration (MIC) determination

MIC was determined in two different ways. For the absorbance assay, MIC was determined as described previously (*Xie et al., 2012*). Briefly, fluconazole was added to liquid YPD medium at a final concentration of 0, 4, 8, 16, 24, 32, 64, or 128 µg/ml in a 96-well plate. A single colony of wild-type or *Cgxbp1Δ* strain was grown in liquid YPD for 16 hr at 30 °C with continuous shaking at 200 rpm. $1\times10^3$ cells were added to each well containing different concentrations of fluconazole. Cell growth was monitored by measuring the absorbance of cell culture at wavelength 600 nm every 15 min for 48 hr in a plate reader (Biotek Cytation3). Each experimental group was performed in duplicates and was repeated three times. For the strip test assay, a single colony of indicated cell strain was grown in 200 µl of liquid YPD. 100 µl of cell suspension was diluted 10 times to measure the optical density at wavelength 600 nm. The cell suspension was diluted with liquid YPD to OD 2.0. A sterile swab was immersed in the diluted cell suspension and subsequently streaked on a YPD agar plate. A MIC strip (Liofilchem, catalog number: 92147) was then placed on the surface of the plate. The plate was incubated at 30 °C for 20 hr before a picture was taken.

## RNAseq analysis

A single colony of *C. glabrata* was cultured for 16 hr in 10 mL of YPD liquid medium at 30 °C with shaking at 200 rpm. The overnight culture was diluted to OD 0.0003 in two separate conditions: (1) 30 mL of YPD for the non-treated group, and (2) 30 mL of YPD containing 64 µg/mL of fluconazole for the fluconazole treatment group. Cultures were grown at 30 °C with shaking at 200 rpm. Cells were harvested when the OD reached 0.6 and subsequently washed twice with ice-cold 1 X TBS buffer. 500 µL of TRIzol (Invitrogen, catalog number: 15596018) was added to the samples before storing at –80 °C. Samples were sent on dry ice to NovoGene Corporation Inc for RNA extraction, library construction, and high-throughput sequencing. The non-directional mRNA libraries were constructed using the NEBNext Ultra RNA Library Prep Kit for Illumina and later sequenced with PE150 on the NovaSeq 6000 platform. Raw reads were aligned to *C. glabrata* reference genome CBS138_s02-m07-r06 using HISAT2 (*Kim et al., 2019*). The read count data were obtained by using featureCounts (version 2.0.3 *Liao et al., 2014*), and normalized expression levels are expressed in FPKM (e.g. fragments on CDS/[mapped reads/10^6×CDS Length/10^3]). These count data were used to determine the differentially expressed genes (DEGs) with DEseq2 (*Love et al., 2014*). Genes with *p*-values less than 0.05 and fold-change greater than 1.5 or less than –1.5 were considered differentially expressed. GO analysis of DEGs was conducted on the FungiExpresZ tool (*Parsania et al., 2023*). The analysis outputs of GO enrichment, correlation, and heatmap were obtained from FungiExpresZ. The volcano plots and dot plots were generated using the online platform SRplot (https://www.bioinformatics.com.cn) (*Tang et al., 2023*).

## Protein extraction and western blotting

For protein extraction from macrophage-internalized *C. glabrata* cells, THP-1 macrophages were infected as described above. At the indicated time post-infection, macrophages were lysed in sterile chilled water, and phagocytosed *C. glabrata* cells were recovered and washed with 1 X TBS buffer, transferred into 1.5 ml microcentrifuge tubes, and stored at –80 °C until use. *C. glabrata* cell pellets were resuspended in 1 X lysis buffer (50 mM HEPES, pH 7.5; 200 mM NaOAc, pH 7.5; 1 mM EDTA, 1 mM EGTA, 5 mM MgOAc, 5% Glycerol, 0.25% NP-40, 3 mM DTT and, 1 mM PMSF) supplemented with protease inhibitor cocktail (Roche). Zirconium beads equivalent to 100 µL volume was added in microcentrifuge tubes and resuspended cells were lysed by six rounds of bead beating on a bullet blender. The sample was centrifuged at 12,000 g at 4 °C for 10 min. Supernatant was carefully transferred to a new tube, and the resultant protein sample was quantified using a Biorad protein assay kit (DC protein assay kit, cat. no. 5000116), and stored in a –80 °C freezer. For western analysis, 25 µg of protein samples were resolved on 12% SDS-PAGE gel and blotted on a methanol-activated PVDF membrane (350 mA, 75 min in a cold room). PVDF membrane was transferred to 5% fat-free milk prepared in 1 X TBST for blocking and incubated for 1 hr. Membranes were probed with appropriate primary (anti-c-MYC antibody, Santa Cruz, cat. no. 9E10 and anti-Histone H3 antibody, Abcam, cat.

no. ab1791) and secondary (goat anti-mouse IgG, Merck Millipore, cat. no. AP124P) antibodies, and Blots were developed by chemiluminescence based ECL western detection kit (GE Healthcare, cat. no. RPN2236) on Chemidoc gel imaging system.

## Acknowledgements

We thank members of the Wong laboratory for their valuable comments throughout the study. We acknowledge the services and technical support from the Genomics and Single Cell Analysis Core and the Drug and Development Core of the Faculty of Health Sciences at the University of Macau. This work was performed in part at the High-Performance Computing Cluster (HPCC), which is supported by the Information and Communication Technology Office (ICTO) of the University of Macau. We thank Lakhansing Pardeshi and Zhengqiang Miao, for Bioinformatics support and Jacky Chan for technical support on the HPC. This work was supported by the Research Services and Knowledge Transfer Office of the University of Macau (Grant number: MYRG2019-00099-FHS and MYRG2022-00107-FHS) and the Science and Technology Development Fund of Macau SAR (FDCT) (Grant number: 0033/2021 / A1 and 0099/2022 /A2).

## Additional information

### Funding

| Funder | Grant reference number | Author |
|---|---|---|
| Research Services and Knowledge Transfer Office, University of Macau | MYRG2019-00099-FHS | Koon Ho Wong |
| Fundo para o Desenvolvimento das Ciências e da Tecnologia | 0033/2021/A1 | Koon Ho Wong |
| Research Services and Knowledge Transfer Office, University of Macau | MYRG2022-00107-FHS | Koon Ho Wong |
| Fundo para o Desenvolvimento das Ciências e da Tecnologia | 0099/2022/A2 | Koon Ho Wong |

The funders had no role in study design, data collection and interpretation, or the decision to submit the work for publication.

### Author contributions

Maruti Nandan Rai, Conceptualization, Data curation, Formal analysis, Validation, Investigation, Writing – original draft, Methodology; Qing Lan, Conceptualization, Data curation, Formal analysis, Investigation, Methodology, Writing - review and editing; Chirag Parsania, Software, Formal analysis, Visualization; Rikky Rai, Investigation; Niranjan Shirgaonkar, Formal analysis, Investigation; Ruiwen Chen, Li Shen, Data curation; Kaeling Tan, Resources, Methodology; Koon Ho Wong, Conceptualization, Resources, Formal analysis, Supervision, Funding acquisition, Validation, Investigation, Visualization, Methodology, Writing – original draft, Project administration, Writing - review and editing

### Author ORCIDs

Maruti Nandan Rai ⬤ http://orcid.org/0000-0002-3430-9263
Qing Lan ⬤ http://orcid.org/0009-0008-3978-7281
Chirag Parsania ⬤ http://orcid.org/0000-0003-4873-2385
Rikky Rai ⬤ http://orcid.org/0000-0002-8786-5650
Niranjan Shirgaonkar ⬤ https://orcid.org/0000-0001-6078-1409
Ruiwen Chen ⬤ http://orcid.org/0000-0001-8780-1078
Koon Ho Wong ⬤ https://orcid.org/0000-0002-9264-5118

### Decision letter and Author response

Decision letter https://doi.org/10.7554/eLife.73832.sa1

Author response https://doi.org/10.7554/eLife.73832.sa2

## Additional files

### Supplementary files

• Supplementary file 1. List of actively transcribing genes in wild-type *C. glabrata* upon macrophage infection.

• Supplementary file 2. List of gene ontology (GO)-terms enriched from temporally induced genes in wild-type *C. glabrata* in response to macrophage infection.

• Supplementary file 3. Lists of iron response genes in wild-type *C. glabrata* during macrophage infection.

• Supplementary file 4. Gene regulatory associations between indicated transcription factors (TFs) and the macrophage infection-induced genes reported in the PathoYeastract database.

• Supplementary file 5. List of orthologues for the macrophage infection-induced transcription factor (TF) and non-TF genes of *C. glabrata* previously shown to have a regulatory association with Xbp1 or Hap3 in *S. cerevisiae* obtained from the PathoYeastract database.

• Supplementary file 6. CgXbp1$^{MYC}$ binding sites upon macrophage infection identified in biological replicates by MACS2 (Model-based Analyses for ChIP-seq) peak-calling.

• Supplementary file 7. List of *C. glabrata* genes having CgXbp1$^{MYC}$ binding sites in their promoters upon macrophage infection.

• Supplementary file 8. List of enriched gene ontology (GO)-terms for biological processes from CgXbp1 targets upon macrophage infection.

• Supplementary file 9. List of transcription factors with CgXbp1 binding at their promoters during macrophage infection.

• Supplementary file 10. List of actively transcribing genes in *Cgxbp1Δ* mutant upon macrophage infection.

• Supplementary file 11. List of gene ontology (GO)-terms enriched from temporally induced genes in *Cgxbp1Δ* in response to macrophage infection.

• Supplementary file 12. Summarized tables of DEGs for wild-type and *Cgxbp1Δ* mutant upon fluconazole treatment.

• Supplementary file 13. List of enriched gene ontology (GO)-terms for biological processes in *Cgxbp1Δ* mutant compared to wild-type *C. glabrata* cells.

• MDAR checklist

### Data availability

RNAPII ChIP-seq, CgXbp1MYC ChIP-seq and RNAseq data are available from the NCBI SRA database under the accession number PRJNA665114, PRJNA743592 and PRJNA1162247, respectively.

The following datasets were generated:

| Author(s) | Year | Dataset title | Dataset URL | Database and Identifier |
|---|---|---|---|---|
| Rai MN, Lan Q, Wong KH | 2020 | Candida glabrata infection of THP1 macrophages | https://www.ncbi.nlm.nih.gov/bioproject/?term=PRJNA665114 | NCBI BioProject, PRJNA665114 |
| Rai MN, Lan Q, Wong KH | 2021 | ChIPseq analysis of Xbp1 in Candida glabrata | https://www.ncbi.nlm.nih.gov/bioproject/?term=PRJNA743592 | NCBI BioProject, PRJNA743592 |
| Rai MN, Lan Q, Wong KH | 2020 | Candida glabrata infection of THP1 macrophages | https://www.ncbi.nlm.nih.gov/bioproject/?term=PRJNA1162247 | NCBI BioProject, PRJNA1162247 |

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
