## [Editor Report]

This important study reveals, with exquisite temporal resolutions, critical transcriptional events that take place as Candida glabrata infects macrophages, providing convincing analyses that enhance our current understanding of the underlying sequential transcriptional changes, including a previously uncharacterized transcription factor (CgXbp1), which plays an important role in modulating the temporal responses in macrophages, impacting C. glabrata survival and virulence and, notably, also fluconazole resistance. The work would benefit from additional experiments that could provide a more mechanistic understanding of the key events leading to successful infection yet, in its current form it should be of interest to a broad audience interested in host-pathogen interactions, fungal biology, and transcriptional mechanisms at large.

---

## [Decision Letter]

**Decision letter after peer review:**

Thank you for submitting your article "Temporal transcriptional response of *Candida glabrata* during macrophage infection reveals a multifaceted transcriptional regulator CgXbp1 important for macrophage response and drug resistance" for consideration by *eLife*. Your article has been reviewed by 4 peer reviewers, and the evaluation has been overseen by a Reviewing Editor and Kevin Struhl as the Senior Editor. The reviewers have opted to remain anonymous.

Essential revisions (for the authors):

- While the ChIP-seq on Xbp1 is interesting, it is obtained in a condition different from the macrophage infection and makes both datasets difficult to compare, as pointed by three of the reviewers. As commented by one of them, information obtained in quiescent cells can not be easily translated to the highly dynamic macrophage environment. Perhaps this is one of the major technical issues of the work as the comparisons of the datasets may (or may not) be yielding relevant information. Potential ways to solve this would be to (i) successfully repeat the Xbp1 ChipSeq analyses in macrophages or (ii) obtain PolII-ChipSeq data from quiescent cells. Of course, the first one is the preferred one as it would really help to elucidate the role of Xbp1 during early times of infection.

A plausible reason of why the authors obtained little correlation in such Chip experiments is that Xbp1 levels are rather low and therefore hard to analyze, which could (indirectly) suggest that Xbp1 may not have an important role in this process. This should be addressed/discussed.

- The relevance of the identified DNA motifs should be further analyzed, particularly as one of them appears quite different from what has been reported in yeast (which could be addressed by having proper PolII data of equivalent datasets, or experimental validation of the motifs through EMSA, DNA footprinting or reporter systems)

- As indicated by the reviewers it is important to better assess the relevance and real significance of the observed fluconazole resistance: i.e MIC, the strength of the phenotype, etc.

- It is also suggested to strengthen some of the conclusions derived from the gene expression data with some experimental validations (i.e at 30 minutes, are C. glabrata actually internalized or just associated?, which may explain the difference in adherence genes at early time points). The paper contains interesting datasets that could provide hints of relevant biological events. It becomes important to explicitly distinguish which are suggested mechanisms (only inferred from expression signatures) to likely mechanisms (combining expression data with data that could help validate such ideas)

There are several other issues pointed out that could be addressed by modifying/editing the text (i.e including relevant references, indicating the new lessons emerging from the dataset, compared with existing microarray datasets, better explaining cut-off values) and that should not require additional experiments.

*Reviewer #1 (Recommendations for the authors):*

While the datasets are valuable and several observations are interesting, it is important to be cautious as the direct targets of CgXbp1 were characterized under one particular condition and the transcriptional analyses were obtained in another condition, one shown to be highly dynamic. Therefore, several inferred targets may or may not be under CgXbp1 control during macrophage infection. Most importantly, as it is, the study does not provide a clear parallel between one list of genes and the other one, to get a glimpse of such concepts. Since CgXbp1 shows to recognize distinct binding motifs, it becomes relevant to understand whether one group behaves differently from the other one in the absence of CgXbp1.

1. Line 180: "similar number of genes were transcribed in the mutant during macrophage infection (1,471 versus 1,589 genes in Cgxbp1Δ and wildtype, respectively) (Supplementary File 5) and ~90% of the transcribed genes are common between wildtype and the mutant (Figure 2—figure supplement 1C), suggesting that CgXbp1 has little effect on the overall set of genes transcribed during macrophage infection"

A relevant question that emerges here, is which are the genes that fail to appear activated in the CgXbp1 mutant. Such analysis is not clearly described in the Results section.

.- Line 265: "While the TCGAG motif is similar to the consensus recognition sequence of *S. cerevisiae* Xbp1 ([TCGA], Mai and Breeden, 1997)"

Please further compare the obtained sequence with other reported consensus sequences for Xbp1, some of which actually share the entire TCGAG core, see

http://cisbp.ccbr.utoronto.ca/TFreport.php?searchTF=T012464_2.00

3. Line 269: "Interestingly, the two motifs have different occurrence among the target promoters bound by CgXbp1MYC with the STVCN7TCT motif occurring approximately three times more frequent than the TCGAG sequence"

While it is true that the authors are performing their ChIP-seq studies in a condition that is quite different from the ones involved in macrophage invasion, it is important to establish some correlative data regarding how these (potentially) two types of promotors behave.

The ideal experiment would be for them to generate PolII-ChIP-seq data from quiescent cells (or if not then RNAseq data), in order to clearly establish co-regulation patterns among the genes of interest, comparing both WT and CgXbp1 mutant.

In addition, one would expect to detect that the genes allegedly being direct targets of CgXbp1 would show a certain level of co-regulation in the existing PolII-ChipSeq data, particularly the groups exhibiting similar cis-elements.

4. Line 332: "at this immediate stage (0.5 h) relative to the other time points (Group 6 genes in Figures 1CandD), indicating global suppression of gene expression in C. glabrata upon macrophage phagocytosis. A recent study showed that the fungal pathogen Cryptococcus neoformans also down-regulate translation during exposure to oxidative stress and suggested that translation suppression may facilitate the degradation of irrelevant transcripts during stress"

Please notice that the commented strategies imply different mechanisms compared with what the authors observed. Thus, while the authors evidenced decreased overall transcriptional rates (as measured by PolII-ChipSeq), the cited work exemplifies decreased translation which appears to also affect the stability of some mRNAs. Most importantly, the authors are not measuring steady-state levels of transcripts (as would be determined by RNAseq) and therefore for transcripts that exhibit medium to long half-lives, a decrease in transcriptional rates may not be causing a dramatic effect in reduced time scales (as compared with highly unstable transcripts).

5. Line 351: "In addition, the ChIP-seq experiment revealed that CgXbp1 directly binds to the promoter of many TFs including 10 carbon catabolite regulators (Figure 5G, Supplementary File 11), suggesting that CgXbp1 indirectly represses the activation of many gene regulatory networks. This probably explains the delayed activation of the carbon catabolic pathway genes"

a. Herein the authors should acknowledge the limitation of their studies as their Xbp1 chip data was obtained under a particular condition, quite different from the dynamic and multi-stimuli environment of a macrophage. Therefore, the identified targets may (or may not) be relevant when interacting with the macrophage.

b. The authors do not discuss whether these 10 genes appear (i) misregulated (higher expression) in the Xbp1 mutant and (ii) what is their behavior during the time course

6. Line 357: "Our overall findings suggest a regulatory model in which global transcriptional repression is established at the early infection stage to withhold transcriptional activation of certain genes whose functions are only required at later stages (Figure 7)"

While this is an interesting model, it is not straightforward to recognize in the dataset that the Xbp1 targets are indeed showing increased expression in the KO during the early stages of infection.

While Xbp1 binding to promoters is an important observation that strongly suggests that such target genes will be subjected to its repressive effect, it can also occur that some targets may not exhibit major changes upon Xbp1 deletion, It is key that the authors compare their Xbp1 Chipseq dataset with the transcriptional data (Pol II for both WT and mutant). As indicated earlier the most straightforward comparison would be to compare Chip and transcriptomic datasets obtained under the same experimental condition.

7. Line 380: "Interestingly, the latter motif (STVCN7TCT) was found at a higher frequency (~3 fold) than the common TCGAG motif from the CgXbp1MYC binding sites, suggesting that CgXbp1 can also form a dimer with another transcription factor that recognizes the STVCN7TCT sequence and that this hetero-dimer controls a larger number of genes than by CgXbp1 alone"

This is an interesting observation and raises the valid question of whether the cohort of genes differing in the type of cis-element present in their promoter show different transcriptional profiles regulated by Xbp1.

8. The discussion does not analyze the reduced virulence observed in Galleria mellonella.

*Reviewer #2 (Recommendations for the authors):*

This manuscript describes the temporal transcriptional response of Candida glabrata during macrophage infection and characterizes the role of the transcriptional repressor CgXbp1 the process. The manuscript is well written, the experiments were well conducted and the subject is very interesting.

However, a few issues should be addressed to improve the quality of the manuscript.

Lines 241-244 – It's difficult to understand the author's justification for failing to obtain reliable ChIP-seq results for Xbp1, when they got them for RNA PolII in the same "ever changing macrophage microenvironment during macrophage infection". The option for defined media makes it difficult to compare with the RNA PolII dataset. Please discuss this issue more thoroughly and, eventually, try again to obtain reliable ChIP-seq results for Xbp1 during macrophage infection.

Line 263 – the two "over-represented motifs" are very different from one another, making it hard to believe that they are both functional. I believe that some demonstration (SPR, EMSA, DNA footprinting, or even something simpler as assessing the effect of promoter mutations in Xbp1 effect on reporter gene expression) of which one works, would be really an important addition to the manuscript.

Line 285 – This section lacks standard MIC determination, to have a clear notion on the impact on fluconazole resistance. Also, the biphasic nature of the fluconazole growth curve is highly unusual. CFU determination conducted along the growth curve would help to assess whether the initial OD variation corresponds to real cell duplication or just changes in cell volume or aggregation.

*Reviewer #3 (Recommendations for the authors):*

The authors should include additional information on how the relative fold-change was calculated, and how the Z-score was determined. Without this information, it is hard to determine whether the upregulation is specific to macrophages, media change, temperature, etc., and therefore the comparator should be clearly defined.

The in-house script should be made available (either methods or github link)

Line 81-83, the way that it is written obscures the fact that 70% of the genes were not bound by PolII during infection. What does this mean for the ability of this technique to identify lowly transcribed genes that may nonetheless play important roles in biology?

---

## [Author Response]

Essential revisions (for the authors):- While the ChIP-seq on Xbp1 is interesting, it is obtained in a condition different from the macrophage infection and makes both datasets difficult to compare, as pointed by three of the reviewers. As commented by one of them, information obtained in quiescent cells can not be easily translated to the highly dynamic macrophage environment. Perhaps this is one of the major technical issues of the work as the comparisons of the datasets may (or may not) be yielding relevant information. Potential ways to solve this would be to (i) successfully repeat the Xbp1 ChipSeq analyses in macrophages or (ii) obtain PolII-ChipSeq data from quiescent cells. Of course, the first one is the preferred one as it would really help to elucidate the role of Xbp1 during early times of infection.A plausible reason of why the authors obtained little correlation in such Chip experiments is that Xbp1 levels are rather low and therefore hard to analyze, which could (indirectly) suggest that Xbp1 may not have an important role in this process. This should be addressed/discussed.

We agree with the reviewers’ concern about comparing datasets of two different conditions. We think that our failure in obtaining biological repeats is a technical difficulty, because the number of fungal cells (and therefore fungal chromatin material) in the infected macrophage samples is limiting, making the immune-precipitation step more difficult than regular ChIP experiments (e.g. the quiescence Xbp1 ChIP experiment).

Nonetheless, we have now successfully repeated the Xbp1 ChIP-seq analysis under macrophage infection condition. The result is now described in the revised manuscript.

- The relevance of the identified DNA motifs should be further analyzed, particularly as one of them appears quite different from what has been reported in yeast (which could be addressed by having proper PolII data of equivalent datasets, or experimental validation of the motifs through EMSA, DNA footprinting or reporter systems)

With the addition of the ChIP-seq data of CgXbp1^MYC^ in the macrophage infection condition and other results, we feel that the manuscript became unfocused and, thus, decided to restructure the paper to mainly focus on CgXbp1 functions during the macrophage infection process. For this reason, the results of the previous motif analysis as well as CgXbp1 function during quiescence have been taken away and will be described in a separate manuscript. We strongly feel that the revised flow of the manuscript can put better emphasis on macrophage infection and drug resistance, which are the main focuses of this work.

- As indicated by the reviewers it is important to better assess the relevance and real significance of the observed fluconazole resistance: i.e MIC, the strength of the phenotype, etc.

We have now performed different ways (e.g. growth curve by absorbance and Liofilchem MIC Test Strips) to determine the fluconazole resistance of wildtype and Cgxbp1Δ mutant cells.

- It is also suggested to strengthen some of the conclusions derived from the gene expression data with some experimental validations (i.e at 30 minutes, are C. glabrata actually internalized or just associated?, which may explain the difference in adherence genes at early time points). The paper contains interesting datasets that could provide hints of relevant biological events. It becomes important to explicitly distinguish which are suggested mechanisms (only inferred from expression signatures) to likely mechanisms (combining expression data with data that could help validate such ideas)

We have done several things to address the relevant comments such as:

Phenotypic plate tests to validate the bioinformatics results;Additional tests to determine drug resistance;Expression analysis of drug transporter genes;Western blot analysis of Xbp1 under different stress conditions;Gene expression analysis of Xbp1 target transcription factors;

For the question about whether C. glabrata cells are actually internalized or just associated, previous reports have shown that C. glabrata cells are successfully phagocytosed within 10-30 min of exposure to macrophages (Roetzer et al., 2010; Seider et al., 2011; Kasper et al., 2014). This is the basis of our design for the time course experiment. Importantly, multiple observations from our gene expression data also indicate that C. glabrata cells are indeed macrophage-internalized at the 30 min time point. For example, we observed at the 30 min time point induction of genes associated with response to oxidative stress, DNA damage repair, autophagy and nutrient deprivation (Figure 1 —figure supplement 3), which are responses to stresses expected after macrophage internalization. The induced expression of the adherence genes is consistent with the infection process, as these genes are known to be required for adherence to macrophages (Katsipoulaki et al., 2024, PMID: 38832801), which is the very first step in the establishment of infection. Hence, it is expected that they are induced at the earliest stage. Considering these together, we think that the observed gene expression profile represents the early responses of macrophage-internalized C. glabrata cells.

In addition, we felt a need to better understand the role of CgXbp1 on fluconazole resistance and hence performed additional RNAseq experiments (even though the reviewers did not request them). The results and a model about CgXbp1 role on fluconazole have been added to the revised manuscript.

There are several other issues pointed out that could be addressed by modifying/editing the text (i.e including relevant references, indicating the new lessons emerging from the dataset, compared with existing microarray datasets, better explaining cut-off values) and that should not require additional experiments.

We have addressed the comments raised by reviewers. Please see the point-by-point response for details.

Reviewer #1 (Recommendations for the authors):While the datasets are valuable and several observations are interesting, it is important to be cautious as the direct targets of CgXbp1 were characterized under one particular condition and the transcriptional analyses were obtained in another condition, one shown to be highly dynamic. Therefore, several inferred targets may or may not be under CgXbp1 control during macrophage infection. Most importantly, as it is, the study does not provide a clear parallel between one list of genes and the other one, to get a glimpse of such concepts. Since CgXbp1 shows to recognize distinct binding motifs, it becomes relevant to understand whether one group behaves differently from the other one in the absence of CgXbp1.

We thank this reviewer for his positive comments and agree with the issues related to non-parallel datasets and gene lists from different conditions. We have now successfully repeated the ChIP-seq experiment of CgXbp1 in the macrophage infection condition (i.e. have matching conditions for both CgXbp1 binding and transcription profiles). With this result, we have decided to rewrite the manuscript focusing on the macrophage infection process and removed the parts about Xbp1’s function in quiescent cells. The comparison between the different motifs identified from Xbp1 binding sites under the two conditions is also taken out from the revised manuscript. We believe that the new flow in the revised manuscript provides a more coherent picture of Xbp1’s function during the early macrophage infection process.

1. Line 180: "similar number of genes were transcribed in the mutant during macrophage infection (1,471 versus 1,589 genes in Cgxbp1Δ and wildtype, respectively) (Supplementary File 5) and ~90% of the transcribed genes are common between wildtype and the mutant (Figure 2—figure supplement 1C), suggesting that CgXbp1 has little effect on the overall set of genes transcribed during macrophage infection"A relevant question that emerges here, is which are the genes that fail to appear activated in the CgXbp1 mutant. Such analysis is not clearly described in the Results section.

The information is now presented in the revised manuscript. There were 295 genes with detectable transcription only in wildtype C. glabrata cells but not in CgXbp1Δ mutant during macrophage infection. Their names and enriched functions are now presented in Supplementary Figure 8a and Supplementary Table 9. The results are described on lines 237-239 in the revised manuscript as follows “Nevertheless, there are 295 and 177 genes with detectable transcription only in wildtype or the Cgxbp1∆ mutant, respectively (Figure 3—figure supplement 1A, Supplementary Table 9).”.

2. Line 265: "While the TCGAG motif is similar to the consensus recognition sequence of S. cerevisiae Xbp1 ([TCGA], Mai and Breeden, 1997)"Please further compare the obtained sequence with other reported consensus sequences for Xbp1, some of which actually share the entire TCGAG core, seehttp://cisbp.ccbr.utoronto.ca/TFreport.php?searchTF=T012464_2.00

As mentioned above, we have now removed the motif result from the revised manuscript to focus on the infection process.

3. Line 269: "Interestingly, the two motifs have different occurrence among the target promoters bound by CgXbp1MYC with the STVCN7TCT motif occurring approximately three times more frequent than the TCGAG sequence"While it is true that the authors are performing their ChIP-seq studies in a condition that is quite different from the ones involved in macrophage invasion, it is important to establish some correlative data regarding how these (potentially) two types of promotors behave.The ideal experiment would be for them to generate PolII-ChIP-seq data from quiescent cells (or if not then RNAseq data), in order to clearly establish co-regulation patterns among the genes of interest, comparing both WT and CgXbp1 mutant.In addition, one would expect to detect that the genes allegedly being direct targets of CgXbp1 would show a certain level of co-regulation in the existing PolII-ChipSeq data, particularly the groups exhibiting similar cis-elements.

As mentioned, we have taken out the results of quiescent experiment and the motif comparison from the revised manuscript.

The suggested co-regulation analysis is a good idea, although we like to note that the expression of CgXbp1 target genes depends on their transcriptional activators, whose activity would be differently controlled during macrophage infection. Therefore, the target genes (if controlled by different activators) would not necessarily have the same expression pattern or might be not expressed at all if their activator is not functional under the experimental condition. Nevertheless, we agree that this is a great suggestion. We have done this for the Xbp1 targets identified from the new data under macrophage infection. The result is presented in Figure 2F and lines 220-227 as follows: “Notably, more than half of the CgXbp1-bound genes (130 out of 220) were minimally transcribed (i.e. they have background levels of RNAPII ChIP-seq signal), if any, during macrophage infection (Figure 2f), presumably their transcription activators are not expressed or functional under the condition. Most of the remaining genes had low expression in wildtype C. glabrata during the early stage of macrophage infection when CgXbp1 expression is at the highest level, while their expression was temporally induced subsequently (Group 2 in Figure 2F), suggesting that CgXbp1 represses their expression during the early infection stage.”.

4. Line 332: "at this immediate stage (0.5 h) relative to the other time points (Group 6 genes in Figures 1CandD), indicating global suppression of gene expression in C. glabrata upon macrophage phagocytosis. A recent study showed that the fungal pathogen Cryptococcus neoformans also down-regulate translation during exposure to oxidative stress and suggested that translation suppression may facilitate the degradation of irrelevant transcripts during stress"Please notice that the commented strategies imply different mechanisms compared with what the authors observed. Thus, while the authors evidenced decreased overall transcriptional rates (as measured by PolII-ChipSeq), the cited work exemplifies decreased translation which appears to also affect the stability of some mRNAs. Most importantly, the authors are not measuring steady-state levels of transcripts (as would be determined by RNAseq) and therefore for transcripts that exhibit medium to long half-lives, a decrease in transcriptional rates may not be causing a dramatic effect in reduced time scales (as compared with highly unstable transcripts).

We agree with this reviewer that the commented strategies are different from what we observed. The sentence has been modified (lines 379-386) to “In addition, gene expression and translation-related genes show the lowest transcription levels (i.e., RNAPII occupancy) at this immediate stage (0.5 h) relative to the other time points (Group 6 genes in Figure 1c,d), indicating global suppression of gene expression in C. glabrata upon macrophage phagocytosis. A recent study showed that the fungal pathogen Cryptococcus neoformans also down-regulates translation during exposure to oxidative stress (Leipheimer et al., 2019). The global suppression of gene expression under stressful conditions probably helps pathogens to reserve energy and resources for coping stress such as the hostile, nutrient-limiting macrophage environment.”

5. Line 351: "In addition, the ChIP-seq experiment revealed that CgXbp1 directly binds to the promoter of many TFs including 10 carbon catabolite regulators (Figure 5G, Supplementary File 11), suggesting that CgXbp1 indirectly represses the activation of many gene regulatory networks. This probably explains the delayed activation of the carbon catabolic pathway genes"a. Herein the authors should acknowledge the limitation of their studies as their Xbp1 chip data was obtained under a particular condition, quite different from the dynamic and multi-stimuli environment of a macrophage. Therefore, the identified targets may (or may not) be relevant when interacting with the macrophage.

As mentioned, we have now gotten the result for CgXbp1 during macrophage infection and rewrote this part according to the new result.

b. The authors do not discuss whether these 10 genes appear (i) misregulated (higher expression) in the Xbp1 mutant and (ii) what is their behavior during the time course

Indeed, there are differences in the binding targets of CgXbp1 under macrophage infection and quiescent, and so the number of carbon regulators also changed from 10 to 4 (Supplementary table 8).

Their expression patterns during infection and in xbp1∆ are now described in Figure 2F and in lines 217, 220-227.

6. Line 357: "Our overall findings suggest a regulatory model in which global transcriptional repression is established at the early infection stage to withhold transcriptional activation of certain genes whose functions are only required at later stages (Figure 7)"While this is an interesting model, it is not straightforward to recognize in the dataset that the Xbp1 targets are indeed showing increased expression in the KO during the early stages of infection.While Xbp1 binding to promoters is an important observation that strongly suggests that such target genes will be subjected to its repressive effect, it can also occur that some targets may not exhibit major changes upon Xbp1 deletion, It is key that the authors compare their Xbp1 Chipseq dataset with the transcriptional data (Pol II for both WT and mutant). As indicated earlier the most straightforward comparison would be to compare Chip and transcriptomic datasets obtained under the same experimental condition.

The expression pattern of identified CgXbp1 targets during macrophage infection has been added to Figure 2F and supplementary table 6. A comparison of RNA pol II ChIP-seq data between wildtype and CgXbp1Δ mutant upon macrophage infection, demonstrated increased expression of ~48% CgXbp1 targets in the CgXbp1Δ mutant. The result is described on lines 194-196 as follows: “The peaks were located at the promoter of 220 genes (Figure 2D, Supplementary Table 6), of which 48% of them were up-regulated during macrophage infection.”.

7. Line 380: "Interestingly, the latter motif (STVCN7TCT) was found at a higher frequency (~3 fold) than the common TCGAG motif from the CgXbp1MYC binding sites, suggesting that CgXbp1 can also form a dimer with another transcription factor that recognizes the STVCN7TCT sequence and that this hetero-dimer controls a larger number of genes than by CgXbp1 alone"This is an interesting observation and raises the valid question of whether the cohort of genes differing in the type of cis-element present in their promoter show different transcriptional profiles regulated by Xbp1.

We agree with this reviewer that the observation is interesting and may inform about differential regulations of CgXbp1 target genes between macrophage infection and quiescence. However, we strongly feel that the results would diverge the focus of this manuscript. Hence, we have decided to remove the motif analysis from the revised manuscript and will report the comparisons of Xbp1 bindings under different conditions in a follow up paper.

8. The discussion does not analyze the reduced virulence observed in Galleria mellonella.

We have added a statement about the reduced virulence phenotype in the Galleria infection model in lines 450-454 of the revised manuscript as the follows: “This suggests that timely and coordinated expression of virulence genes is crucial for C. glabrata’s survival and pathogenic response during macrophage infection. Presumably, the pathogen needs to strategize the utilization of cellular resources to survive and counteract host attacks during infection, and this may also be the reason for the reduced virulence in the Galleria infection model.”

Reviewer #2 (Recommendations for the authors):This manuscript describes the temporal transcriptional response of Candida glabrata during macrophage infection and characterizes the role of the transcriptional repressor CgXbp1 the process. The manuscript is well written, the experiments were well conducted and the subject is very interesting.

We thank this reviewer for the positive remarks.

However, a few issues should be addressed to improve the quality of the manuscript.Lines 241-244 – It's difficult to understand the author's justification for failing to obtain reliable ChIP-seq results for Xbp1, when they got them for RNA PolII in the same "ever changing macrophage microenvironment during macrophage infection". The option for defined media makes it difficult to compare with the RNA PolII dataset. Please discuss this issue more thoroughly and, eventually, try again to obtain reliable ChIP-seq results for Xbp1 during macrophage infection.

We believed that was a technical difficulty, which we have now overcome. The result of ChIP-seq data of Xbp1 during macrophage infection is now included and described in lines 182-227. As a result of this addition, we have restructured the manuscript to focus on macrophage infection and removed all the data related to quiescent.

Line 263 – the two "over-represented motifs" are very different from one another, making it hard to believe that they are both functional. I believe that some demonstration (SPR, EMSA, DNA footprinting, or even something simpler as assessing the effect of promoter mutations in Xbp1 effect on reporter gene expression) of which one works, would be really an important addition to the manuscript.

As mentioned, we have now rewritten the manuscript to focus on the infection process. As a result, the motif analysis results are removed from the revised manuscript.

Line 285 – This section lacks standard MIC determination, to have a clear notion on the impact on fluconazole resistance. Also, the biphasic nature of the fluconazole growth curve is highly unusual. CFU determination conducted along the growth curve would help to assess whether the initial OD variation corresponds to real cell duplication or just changes in cell volume or aggregation.

As suggested, we have now determined the MIC of wildtype and Cgxbp1∆ mutant using growth analysis and MIC strips. The results are presented in Figure 5B-F and lines 301-312.

Reviewer #3 (Recommendations for the authors):The authors should include additional information on how the relative fold-change was calculated, and how the Z-score was determined. Without this information, it is hard to determine whether the upregulation is specific to macrophages, media change, temperature, etc., and therefore the comparator should be clearly defined.

The information is now provided in lines 556-560 as follows: “Fold changes for the time course experiment were calculated with respect to 0.5 h, while fold changes for DEGs are relative to WT (i.e. ∆ / WT). For the z-score plots, only genes whose expression changes at least two folds between any two or more time points during the macrophage infection experiment were included. Z-scores across the time points were generated using the row clustering option in FungiExpresZ (Parsania et al., 2023).”.

The in-house script should be made available (either methods or github link)

The scripts are now available on github and the github links (https://github.com/RaimenChan/Xbp1_project and https://github.com/zqmiao-mzq/perl_tools/blob/master/zqWinSGR-v4.pl) are given in the text in lines 541 and 581.

Line 81-83, the way that it is written obscures the fact that 70% of the genes were not bound by PolII during infection. What does this mean for the ability of this technique to identify lowly transcribed genes that may nonetheless play important roles in biology?

While the PolII ChIP-seq technique is powerful in detecting major transcription events and changes, it does indeed have a limitation in detecting very lowly transcribed genes. As a result, responses of lowly transcribed genes during macrophage infection would have been missed from this study. A statement about this limitation has been added to the Discussion on lines 358-363 as follows: “Through mapping genome-wide RNAPII occupancy, our result reveals that about 30% of C. glabrata genes transcribed during the adaptation, survival, and growth inside the alien macrophage microenvironments. The number of genes may be an underestimate of the overall response, as the RNAPII ChIP-seq method has a narrow, low detection range relative to RNAseq and may not be able to detect lowly transcribed genes. Nevertheless, our data reveal dynamic temporal responses during macrophage infection.”.